# Progressive veining during peridotite carbonation: insights from listvenites in Hole BT1B, Samail ophiolite (Oman)

Manuel D. Menzel[1,2], Janos L. Urai[1], Estibalitz Ukar[3], Thierry Decrausaz[4], Marguerite Godard[4]

[1] Tectonics and Geodynamics, RWTH Aachen University, Lochnerstrasse 4-20, D-52056 Aachen, Germany
[2] now at: Instituto Andaluz de Ciencias de la Tierra, Avenida de Palmeras 4, 18100 Armilla, Spain
[3] University of Texas at Austin, Bureau of Economic Geology, TX, USA
[4] Géosciences Montpellier, CNRS, Université de Montpellier, Montpellier, France

* *Correspondence to*: Manuel D. Menzel (manuel.menzel@csic.es)

**Abstract.** The reaction of serpentinized peridotite with $CO_2$-bearing fluids to form listvenite (quartz-carbonate rock) requires massive fluid flux and significant permeability despite increase in solid volume. Listvenite and serpentinite samples from Hole BT1B of the Oman Drilling Project help to understand mechanisms and feedbacks during vein formation in this process. Samples analyzed in this study contain abundant magnesite veins in closely spaced, parallel sets and younger quartz-rich veins. Cross-cutting relationships suggest that antitaxial, zoned magnesite veins with elongated grains growing from a median zone towards the wall rock are among the earliest structures to form during carbonation of serpentinite. Their bisymmetric chemical zoning of variable Ca and Fe contents, a systematic distribution of $SiO_2$ and Fe-oxide inclusions in these zones, and cross-cutting relations with Fe-oxides and Cr-spinel indicate that they record progress of reaction fronts during replacement of serpentine by carbonate in addition to dilatant vein growth. Euhedral terminations and growth textures of magnesite vein fill together with local dolomite precipitation and voids along the vein-wall rock interface suggest that these veins acted as preferred fluid pathways allowing infiltration of $CO_2$-rich fluids necessary for carbonation to progress. Fracturing and fluid flow was probably further enabled by external tectonic stress, as indicated by closely spaced sets of subparallel carbonate veins. Despite widespread subsequent quartz mineralization in the rock matrix and veins, which most likely caused a reduction in the permeability network, carbonation proceeded to completion within listvenite horizons.

## 1 Introduction

Listvenites (or listwanites) are the result of large-scale fluid-rock interaction that converts serpentinized peridotite into magnesite – quartz rock due to reactive flux of $CO_2$-bearing aqueous fluids. During this reaction, which typically occurs at low temperature hydrothermal to sub-greenschist facies conditions (80 – 350 °C) (Beinlich et al., 2020b; Beinlich et al., 2012; Falk and Kelemen, 2015; Johannes, 1969; Menzel et al., 2018), the rock gains more than 30 wt% $CO_2$ and an equivalent increase in rock volume. Because of the large amount of $CO_2$ that can be captured through this process and the fact that reaction kinetics can be fast, especially if peridotite contains brucite or remnant olivine (Kelemen et al., 2011), listvenites have attracted considerable attention as a natural analogue for carbon sequestration by mineral carbonation (Kelemen et al., 2020d; Matter and Kelemen, 2009; Power et al., 2013). The International Panel of Climate Change (IPCC) predicts that negative carbon emission techniques such as artificially enhanced mineral carbonation will become necessary even if the transition to a fossil carbon-free economy is faster than expected (IPCC, 2021). Unfortunately, although listvenites occur in many ophiolites worldwide (e.g., Boskabadi et al., 2020; Emam and Zoheir, 2013; Hansen et al., 2005; Hinsken et al., 2017; Qiu and Zhu, 2018) and isotopic evidence suggests carbonation of serpentinized peridotite can proceed very fast in nature (Beinlich et al., 2020a), experiments have so far not been able to reproduce this reaction at a large scale and acceptable costs.

Most experimental efforts have focused on reproducing and enhancing the conditions of mineral carbonation related to hyperalkaline springs and subaerial and subaquatic weathering (Kelemen et al., 2020d; Oskierski et al., 2013; Power et al.,

2013). Nonetheless, the conversion of peridotite to listvenite has a higher potential for carbon sequestration in comparatively smaller rock volumes. The reasons why experimental listvenite formation at a large scale is challenging are the comparatively high temperatures (> 80 °C, but ideally about 170 – 200 °C; Kelemen and Matter, 2008), the low permeability of ultramafic rocks, and the volumetric expansion and permeability reduction associated with conversion to magnesite and quartz

(Hövelmann et al., 2013; Peuble et al., 2018; van Noort et al., 2017). Besides the necessity for a source of $CO_2$-rich fluids, permeability reduction is likely one of the main limiting factors controlling the extent of listvenite formation in natural systems. Because natural fluids at sub-greenschist facies conditions are typically not highly enriched in $CO_2$, listvenite formation requires high fluid-rock ratios, which implies that permeability must be maintained or repeatedly renewed over a long period to guarantee continued fluid flux.

An ingredient that is absent in most experiments but is commonplace in nature is tectonic deformation and related deviatoric stress. Listvenites occur suspiciously often along major shear zones in various ophiolites (Ash and Arksey, 1989; Belogub et al., 2017; Menzel et al., 2018; Qiu and Zhu, 2018). In listvenites of the Samail ophiolite, Oman, ductile deformation microstructures demonstrate that deformation and carbonation were synchronous (Menzel et al., 2021), suggesting that tectonic deformation plays a key role in maintaining the permeability necessary for long-lasting reactive fluid flow. Most serpentinites and listvenites formed at T < 200 °C are expected to be brittle, although high fluid pressure can also lead to reaction-assisted

ductile deformation at low temperature (Menzel et al., 2021). Deformation can enhance permeability by fracturing (e.g., Sibson, 1996) and grain-scale processes including dilatant granular flow and creep cavitation (e.g., Fusseis et al., 2009). Different sets of veins filled with carbonate and/or quartz are commonplace in listvenites and related carbonate-bearing serpentinites (Beinlich et al., 2020b; Beinlich et al., 2012; Hansen et al., 2005; Menzel et al., 2018), suggesting that dilatancy

may be one of the main mechanisms by which carbonation of peridotite can progress.
Although early veins have been identified in listvenites of the Samail ophiolite (Beinlich et al., 2020b; Menzel et al., 2021), the role fractures and crystallization pressure played in allowing penetration of $CO_2$-rich fluids and listvenite formation are uncertain. Here we investigate the progression of microstructures within veins formed as a result of carbonation in serpentinites and listvenites of Hole BT1B, Oman Drilling Project (International Continental Drilling Project Expedition 5057- 4B) using

advanced high-resolution imaging and analytical methods. This natural laboratory and unmatched sample quality offer an ideal opportunity to investigate the relative timing and sequence of fracturing and mineralization, their relationship with carbonation of the unfractured matrix, the role of replacement veining, and the evolution of porosity throughout the carbonation processes in order to evaluate whether fracturing helped accommodate volumetric expansion and increased the permeability necessary for the progression of carbonation.

## 2 Geological setting and previous work

Due to the high-quality of the drill cores recovered at site BT1 and the interdisciplinary analysis in the frame of the Oman Drilling Project, the Samail ophiolite hosts the best-studied listvenites worldwide. In the following sections, we briefly summarize the geological setting and the most important results of previous studies on listvenites in this area.

### 2.1 The Samail Ophiolite

The Samail ophiolite (Oman) (Fig. 1) is composed of a sequence of obducted oceanic crust (4 – 8 km thickness) and mantle (up to 12 km thick) with similar characteristics as fast-spreading oceanic crust in the Pacific (Hopson et al., 1981; Nicolas et al., 1996; Rioux et al., 2013). Crystallization of the oceanic crust occurred during the Cenomanian (96.4-95.5 Ma) (e.g., Coleman, 1981; Rioux et al., 2013) in a mid-ocean ridge (Boudier and Coleman, 1981; Hacker et al., 1996) or, most likely, in a supra-subduction zone setting (MacLeod et al., 2013; Pearce et al., 1981; Searle and Cox, 2002). A first stage of hot oceanic

subduction, broadly coeval with or shortly after oceanic crust crystallization (Garber et al., 2020) produced greenschist to

granulite facies metabasalts and minor metasediments, which were underplated as a metamorphic sole below the ophiolite and are now exposed as discontinuous layers and lenses along the basal thrust fault of the ophiolite. Different parts of the metamorphic sole record temperatures of 450 – 550 °C and 700 – 900 °C at 0.7 – 1.2 GPa (Kotowski et al., 2021; Soret et al., 2017). The ophiolite and metamorphic sole were emplaced by top-to-SE thrusting onto allochthonous marine sediments (the

Hawasina nappes). Further shortening subsequently culminated in NE-dipping subduction and burial of the autochthonous Arabian continental margin below the Hawasina nappes and the ophiolite (81 – 77 Ma) (Garber et al., 2021). The deepest burial of the under-thrust margin occurred at about 79 Ma (Warren et al., 2003), reaching peak metamorphic conditions of 280 – 360 °C and 0.3 – 1.0 GPa in rocks exposed in the Jebel Akhdar and Saih Hatat domes and 450 – 550 °C and 2.0 – 2.4 GPa in eclogites at As Sifah (Agard et al., 2010; Grobe et al., 2019; Miller et al., 1999; Saddiqi et al., 2006; Searle et al., 1994).

After termination of obduction, N-S extension during the Maastrichtian and early Paleocene led to top-to-NNE, mostly bedding-parallel shear zones in the autochthonous units (Grobe et al., 2018, and references therein). This was followed by normal faulting, folding and low-angle detachments in the Eocene and strike-slip faulting in the Oligocene related to the exhumation of the Jebel Akhdar and Saih Hatat anticlinoria and tectonic and erosional thinning of the ophiolite (e.g., Grobe et al., 2019; Mattern and Scharf, 2018, and references therein).

**2.2 Listvenites in the Samail Ophiolite**

Listvenites occur along or in close proximity to the basal thrust of the Samail ophiolite at the western contact of the Aswad massif (United Arab Emirates and North-Western Oman) and in the Northern Samail massif (Fanjah region, Oman) (Glennie et al., 1974; Stanger, 1985; Wilde et al., 2002) (Fig. 1a). In the area surrounding OmanDP Site BT1 in the Northern Samail massif, listvenites crop out as 10s-of-meter-thick bands along and parallel to the contact between banded peridotites and the

underlying metamorphic sole, which in turn is underlain by multiply deformed, allochthonous metasediments of the Hawasina nappes (Fig. 1b). Directly north of Site BT1, these units form a broad anticline (Falk and Kelemen, 2015). Early models proposed that listvenite formation was related to sub-meteoric fluids and normal faulting during extensional tectonics in the Paleogene, after ophiolite emplacement (Nasir et al., 2007; Stanger, 1985). However, the geometry of the listvenite outcrops and evidence for ductile deformation synchronous with carbonation (Menzel et al., 2021) indicates that listvenites formed

along a shallow-dipping fault zone at the interface between ophiolite, metamorphic sole, and underlying metasediments, and not along steep normal faults (Kelemen et al., 2022). Together with an imprecise internal Rb-Sr isochron age of 97±29 Ma (2σ) for Cr-muscovite-bearing listvenite close to Site BT1 (Falk and Kelemen, 2015), this points to listvenite formation during subduction/underthrusting of the Arabian continental margin below the obducting ophiolite, consistent with a deep source of $CO_2$-bearing fluids as inferred from Sr and C stable isotope geochemistry (de Obeso et al., 2022). $CO_2$-fluid flux and

carbonation concurrent with an early reactivation of the basal ophiolite thrust fault as an extensional decollement would also be consistent with the outcrop geometry, and extensional top-to-the-NE shearing in the authochtonous carbonates below the ophiolite (64±4 Ma) (Hansman et al., 2018) falls just within the 2σ margin of the Rb-Sr isochron of Falk & Kelemen (2015). However, while possible, based on the currently available data this is less likely than a subduction-obduction setting (Kelemen et al., 2022). The listvenites contain abundant veins of various generations (Fig. 1 c & d) and, together with adjacent units,

have been overprinted by cataclasis and sharp normal to strike-slip faults that obscure the original structures, showing that multiple brittle deformation phases occurred after listvenites formed (Menzel et al., 2020).

**2.3 Serpentinites and listvenites of Hole BT1B**

Hole BT1B consists in its upper part of listvenite intercalated with two serpentinite layers separated by a fault at 200 m downhole depth from underlying greenschist-facies metamafic rocks of the metamorphic sole (Fig. 1e) (Kelemen et al., 2020b).

At Site BT1, magnesite predominates, while dolomite and calcite are common in listvenites further north in the Fanjah area. Clumped isotope thermometry, the presence of quartz-antigorite (± talc) intergrowths, and recrystallization microstructures of

quartz after opal point to listvenite formation temperatures of 80 – 150 °C in this area (Falk and Kelemen, 2015). Estimates of vein and matrix carbonate precipitation in serpentinite and listvenite of core BT1B range from 45±5 to 247±52 °C based on clumped isotope thermometry (Beinlich et al. 2020). The pressure and depth of listvenite formation are less well constrained.

Based on data from underlying carbonate sediments (Grobe et al., 2019) and a plausible ophiolite thickness of 8 – 10 km, pressure was at least ~0.3 GPa, while the P-T conditions recorded by the metamorphic sole set an upper bound of 0.7 – 1.0 GPa (Kotowski et al., 2021; Soret et al., 2017). Except for some dolomite-enriched intervals, especially close to the basal fault, most BT1B listvenites are composed of magnesite, quartz, minor Cr-spinel, and locally Fe-(hydr)oxides or Cr-bearing muscovite (fuchsite) (Kelemen et al., 2020b). The bulk chemistry and proportions of magnesite and quartz can vary

significantly on a small scale (Okazaki et al., 2021), but at the meter scale they are consistent with overall isochemical replacement of peridotite and minor addition of fluid-mobile elements (Godard et al., 2021; Kelemen et al., 2020b). Massive listvenite domains show two main types of pervasive microstructures: (i) zoned, ellipsoidal to spheroidal magnesite particles with euhedral to dendritic habit in a finer-grained quartz matrix (Beinlich et al., 2020b; Menzel et al., 2021), and (ii) variably large quartz (±fuchsite) aggregates with microstructures resembling those of orthopyroxene or bastite surrounded by a matrix

of vermicular, mesh-like magnesite – quartz intergrowths (Kelemen et al., 2020b; Menzel et al., 2021). Trace element geochemistry suggests that the protolith of the BT1B listvenites was part of the banded peridotite unit commonly found at the base of the Samail ophiolite, with compositions of fuchsite-bearing listvenite overlapping with amphibole-bearing basal lherzolite and fuchsite-free listvenite similar to the composition of refractory peridotite (Godard et al., 2021). Sr and C isotope geochemistry points to deep-sourced metamorphic fluids derived from meta-sediments similar to the underlying Hawasina

Formation as the $CO_2$ source (de Obeso et al., 2022).

Visual core logging by the OmanDP science team showed that veins are abundant in serpentinite and listvenite, with densities of > 200 veins per meter of core for veins < 1 mm wide, and 50 – 200 veins per meter for veins > 1 mm (Kelemen et al., 2020b). In serpentinites, the vein logging team distinguished between four main vein types, with a narrow (< 0.1 mm) serpentine vein network that defines a mesh texture being the earliest generation. The serpentine mesh is cut by multiple

generations of serpentine veins, early carbonate–oxide veins characterized by a Fe-oxide-bearing median zone and antitaxial growth habit, and younger carbonate veins with rare quartz (Kelemen et al., 2020b). In listvenites, the vein logging team identified narrow (< 0.1 mm) carbonate-oxide veins with antitaxial habit and a median line as the earliest vein generation, followed by discontinuous carbonate veins, comparatively wider carbonate – chalcedony/quartz veins (> 1 mm to > 1 cm), and late, partially open dolomite and/or magnesite veins (Kelemen et al., 2020b).

Based on deformation and overgrowth microstructures of folded and ductile transposed veins, Menzel et al. (2021) concluded that the early carbonate-oxide veins in listvenites formed during the incipient stage of the carbonation reaction, while most of the rock was still composed of serpentine, confirming similar inferences from previous studies (Beinlich et al., 2020b; Kelemen et al., 2020b). In this study, we refine the preliminary vein classification of the core logging (c.f. Tables 1 & 2) and investigate in detail those vein generations that were directly involved in the carbonation reaction progress in order to understand the

mechanisms controlling focused fluid flux, permeability and reactivity during carbonation.

## 3 Methods and materials

### 3.1 Samples

The highly variable range of (micro)structures in serpentinite and listvenite of Hole BT1B was sampled during the Oman drilling Phase 1 core logging onboard R/V Chikyu in September 2017. A few additional samples from the area north and east

of Hole BT1B were obtained during a field campaign in January 2020. Thin sections produced from core samples are oriented with respect to the core reference frame (CRF), an arbitrary orientation along which contiguous core sections were split. Due to the inclination of 75° of Hole BT1B and discontinuities across which the orientation of core sections could not be

reconstructed, structural measurements and sample orientations are not easily comparable between different parts of the core. Thin sections produced from field samples were either oriented in relation to structural elements, i.e. perpendicular to foliation, or, when no foliation was visible, in the geographical reference frame. For this study, vein microstructures and cross-cutting relationships were inspected in 115 thin sections, of which a subset lacking late cataclastic overprint was investigated in more detail. Our vein classification is primarily based on listvenite samples from Hole BT1B. Thin sections and samples from BT1B are named here with an abbreviated form of the ICDP convention, following the scheme "*Hole*"_"*Core*"-"*Section*"_"*top*"-"*bottom*", where top/bottom denote the distance (in cm) from the top of the section.

**3.2 Optical and scanning electron microscopy**

A PetroScan Virtual Polarizing Microscope (ViP; RWTH Aachen University) was used to obtain high-resolution scans of full thin sections in plane-polarized light, reflected light, and at 10 different crossed-polarizer orientations with a 10x objective. A high-precision automated stage allows the interpolation of the extinction behavior of each pixel to visualize extinction at all polarization angles. A selection of these digitized thin sections are publicly available as Supplementary material of the Oman Drilling Project (Kelemen et al., 2020b) at http://publications.iodp.org/other/Oman/SUPP_MAT/index.html#SUPP_MAT_Z. Of these, samples of special interest regarding vein microstructures are: *Core 14-3, 77-80*; *Core 31-4, 12-14*; *Core 35-1, 30-32*; *Core 44-3, 9-11* and *Core 67-2, 36-40*.

Back-scattered electron (BSE) and energy-dispersive X-ray spectroscopy (EDX) maps were acquired for phase identification and imaging of chemical zoning using a Zeiss Gemini SUPRA 55 field-emission scanning electron microscope (FE-SEM) at the Institute of Tectonics and Geodynamics of RWTH Aachen University. Whole thin sections and specific areas of interest were mapped at an acceleration voltage of 15 kV, 8.5 mm working distance, and dwell times of 0.2 – 1.5 ms/point. Samples were coated with a 6 – 8 nm-thick layer of tungsten for conductivity.

We used a Zeiss Axio Scope optical microscope equipped with a "cold" cathode luminoscope CL8200 MK5-2 to obtain optical-CL panorama images of large thin section areas. Single images were taken at operating conditions of 15 kV, 320 – 350 µA with a 10x objective, and exposure times of 5 – 10 s.

In selected samples and areas of specific interest, panchromatic and blue-filtered SEM-CL images were obtained using a Zeiss Sigma High Vacuum FE-SEM equipped with a Gatan MonoCL4 system at the University of Texas at Austin. Following the guidelines of Ukar and Laubach (2016), carbon-coated thin sections were imaged at 5 kV accelerating voltage, 120 µm aperture, 125 µs dwell time, and 2048 x 2048 pixel resolution at magnifications up to 2500x.

One serpentinite sample with subparallel serpentine veins was prepared with broad ion beam (BIB) polishing to image nano-porosity in matrix and vein serpentine. For this, a 6 x 6 x 4 mm prism was cut from a non-weathered part of the sample, and mechanically pre-polished. BIB polishing was performed in a Leica TIC3X with rotary stage (RWTH Aachen) applying three Ar-ion beams for 6 hours at 3 kV acceleration voltage, a beam current of 1.0 mA for each beam, and an incidence angle of 4.5°. The tungsten-coated sample surface was imaged at high resolution (10 – 20 nm pixel size) in secondary electron (SE) mode using the Zeiss Gemini SUPRA 55 FE-SEM (RWTH Aachen) at an acceleration voltage of 3 kV and a working distance of 5 mm.

**4 Results**

**4.1 Veins in serpentinite**

**4.1.1 Serpentine mesh, magnetite-serpentine veins, and serpentine crack-seal veins**

Networks of serpentine-mesh veins with the typical polygonal hourglass texture of serpentinite (Wicks and Whittaker, 1977) are ubiquitous in non-foliated serpentinites (Fig. 2a; ss0 in Table 1). In these veins, serpentine is usually brownish in transmitted light and magnetite is abundant. The serpentine mesh is cut by various generations of magnetite-serpentine veins

and serpentine crack-seal veins (ss1 & ss2, Table 1; Fig. 2 b & c). Similar to the serpentine mesh, magnetite-serpentine veins contain a median zone rich in magnetite with flaky to fibrous serpentine towards the vein walls. Serpentine is clear in transmitted light and the veins are discrete and continuous, occasionally forming parallel sets. Serpentine crack-seal veins are characterized by uniform to cloudy extinction under crossed polarizers with similar fiber orientations over the entire length of the vein. Commonly, these veins show vein-parallel banding with oscillatory extinction patterns that are typical of serpentine crack-seal veins (Andreani et al., 2004), likely due to alternating precipitation of chrysotile and lizardite during crack-seal cycles (Tarling et al., 2021).

Clusters of parallel, en-echelon and/or branched serpentine veins occur in foliated serpentinites of the area north of site BT1. The veins are parallel to the penetrative serpentinite foliation, which is defined by flattened mesh cells delineated by magnetite aggregates (Fig. 2d). Serpentine in these cleavage-parallel veins has the same extinction direction with the lambda plate, indicating a strong and consistent shape and crystallographic preferred orientation, which is different from the crystallographic preferred orientation of matrix serpentine (Fig. 2b).

### 4.1.2 Pseudomorphic carbonate

In carbonate-bearing serpentinite, small magnesite aggregates locally occur along the serpentine mesh, pseudomorphically tracing the polygonal mesh outlines (Fig. 2 e, f; sc0, Table 1; supplementary Fig. S1). Magnetite, partly transformed to Fe-magnesite, is commonly present as inclusions within these magnesite aggregates. Pseudomorphic magnesite has core-rim zoning of variable Fe contents, and magnesite rims in the vein network commonly have euhedral crystal facets towards contacts with serpentine, or a dendritic habit (Fig. 2g). In places, the pseudomorphic magnesite vein network has a preferred orientation tracing flattened serpentine mesh cells.

Pseudomorphic replacement of serpentine by carbonate also occurs within and along the walls of serpentine crack-seal veins (Fig. 2h). Here, vein-perpendicular magnesite columns locally replace serpentine along serpentine fibers.

### 4.1.3 Carbonate veins

Most carbonate in serpentinites of Hole BT1B is in veins, and only occurs in minor amounts as dispersed grains within the serpentine matrix (Fig. 3). Besides pseudomorphic carbonate, two types of early carbonate veins dominate in serpentinite: patchy to feathery carbonate veins parallel to serpentine cleavage planes, and zoned magnesite veins.

Cleavage-parallel carbonate veins (sc1, Table 1) have an irregular morphology with highly variable vein width and consist of Fe-poor magnesite and/or dolomite (Fig. 3 c, d). These veins follow the (locally folded) serpentine cleavage and often form branched and curved, semi-isolated patches with splayed vein tips.

The most abundant carbonate vein type in serpentinite consists of zoned magnesite veins (sc2, Table 1; Fig. 3 c-e). During core logging, these have been named carbonate-oxide or antitaxial carb-oxy veins (Kelemen et al., 2020b). Because an Fe-oxide median line and antitaxial texture is not always present, we hereafter refer to this vein group as zoned magnesite veins (sc2). These are characterized by a planar morphology with rather constant vein width (typically $50 - 200$ µm), and a vein-parallel, often bisymmetrical chemical zoning from a median zone towards the vein walls (Fig. 4). The veins are mostly composed of magnesite of variable composition, but dolomite can also be present along the vein walls, and minor vein segments locally consist of (or are replaced by) dolomite and rare quartz. The type of zoning and width of different zones varies between different samples, and, occasionally, within the same thin section. Where present, the median line ($2 - 10$ µm wide) consists of Fe-oxide, and/or soft Fe-hydroxides (possibly goethite) that are rarely well preserved during thin section preparation. The median zone is commonly composed of Ca- and Fe- bearing magnesite in the center, followed by Fe-rich magnesite with systematic variations in the number of $SiO_2$-inclusions, and Fe-poor magnesite or dolomite towards the vein walls (Fig. 4 d, f). Chemical zoning is not always well developed and may consist only of slight variations in the amount of silica impurities in magnesite. Flat or lens-shaped serpentine inclusions parallel to the vein length are common. In some zoned

magnesite veins from core sections of the lower part of the upper serpentinite layer (98.7 – 100 m down-hole depth, c.f. Fig. 1e) seams of talc occur along the vein walls, and locally talc is present as remnant inclusions between the Fe-rich and Fe-poor magnesite zones of the veins (Fig. 4 c, f).

In some serpentinite core intervals, sc2 veins form closely spaced, parallel sets (clusters) (Fig. 3; Fig. 4a). Locally, they occur as two conjugate, anastomosing sets that resemble an s-c fabric of scaly fractured serpentinite (Fig. 3 b, e). Where they intersect elongated Cr-spinel grains, zoned magnesite veins branch into numerous narrow veinlets, locally fragmenting the Cr-spinel (Fig. 4b). Elongated/flattened Cr-spinel is commonly aligned, probably showing the orientation of a remnant high-temperature plastic deformation fabric of former olivine. All carbonate veins in Hole BT1B are oblique to that early fabric. No systematic relationship between sc1-sc2 vein orientations and the serpentinite mesh texture is apparent. Cross-cutting relationships between sc2 and sc1 veins are usually ambiguous because zoned magnesite veins are locally deflected along serpentine cleavages. However, sc2 veins locally branch into narrow veinlets where they transect sc1 veins, indicating that at least some sc2 are relatively younger.

Cross-cutting relationships between sc2 veins and mesh-pseudomorphic magnesite aggregates in the serpentinite matrix are also often complex and ambiguous. For example, figure 4g shows that only the median zone of the sc2 vein cuts the matrix magnesite aggregate, while the Fe-enriched magnesite zone (II) does not. Facetted crystal terminations of carbonate towards matrix serpentine are locally present along the walls of some sc2 veins (yellow arrow in Fig. 4g). Sc2 veins commonly pinch out in narrow vein tips, but abrupt, partially corroded, wide vein terminations are also present. In places, magnesite veins show narrow talc vein terminations (Fig. 4h), Fe-oxides veinlets (Fig. 4i), or feathery quartz veins (sq1, see below) that emanate from carbonate vein tips (Fig. 5a).

### 4.1.4 Quartz veins

Quartz veins are common in the serpentinites of Hole BT1B, unlike typical serpentinites and peridotites of the Samail ophiolite where they are mostly absent. Quartz-serpentine intergrowths have previously been observed in samples near Hole BT1B (Falk and Kelemen, 2015) and a few quartz veins were logged during shipboard core description (Kelemen et al., 2020b). We note that their abundance in BT1B was underestimated during logging because they are usually narrow and easily overlooked (Fig. 5a). Two types are common: "feathery" quartz vein aggregates intergrown with serpentine at the micro-scale (sq1; Table 1) and wider, poly-granular to blocky quartz / quartz-magnesite veins (sq2; Table 1). Cross cutting relationships between both types of quartz veins are ambiguous. Sq1 veins are strongly branched, locally emanating from the tips of carbonate veins (Fig. 5a). They commonly cut and may offset sc2 carbonate veins (Fig. 5b). Some wide sq1 veins (> 20 μm) show an antitaxial habit, with a median zone composed of pure quartz and margins enriched in nm- to μm-sized serpentine and/or carbonate inclusions (Fig. 5 c-f). Pods of dolomite or magnesite constitute parts of some veins.

Sq2 quartz / quartz-magnesite veins locally contain euhedral magnesite and are commonly oriented parallel to sc2 veins (Fig. 3d; Fig. 4e), suggesting that they formed due to preferential fracturing along the walls of zoned magnesite veins.

### 4.1.5 Late carbonate veins

Late, partially open or brecciated carbonate veins cut serpentinites and all previous vein generations (sc4, Table 1). These are unrelated to the formation of listvenite, and possibly linked to young magnesite, dolomite and calcite/aragonite precipitation in open joints from groundwater or hyperalkaline serpentinization fluids. Similar young carbonate veins and travertine are common in the weathering horizon of the Samail ophiolite peridotites (e.g., Chavagnac et al., 2013; Giampouras et al., 2020; Noël et al., 2018; Ternieten et al., 2021). Therefore, we do not consider them further here, but we note that they can locally obscure structures of veins synchronous with listvenite formation.

## 4.2 Veins in listvenite

Some listvenite intervals of Hole BT1B are highly veined, such that locally > 50 % of the listvenite volume consists of veins. This is the result of pseudomorphic replacement of previous serpentine veins, and the superposition of veins formed at different time steps (Table 2).

### 4.2.1 Pseudomorphic veins after serpentine

Based on their strong microstructural resemblance with common serpentine veins in the serpentinites, we identify two types of pseudomorphic veins: pseudomorphic magnesite (and/or quartz) vein networks after serpentine mesh ($l_{ss0}$), and magnesite-quartz veins after serpentine crack-seal veins ($l_{ss2}$). Incipient stages of both pseudomorphic vein replacement microstructures are also present in carbonate-bearing serpentinite.

The mesh-pseudomorphic vein network ($l_{ss0}$) is most evident in listvenites where the volume in between the magnesite mesh consists of monomineralic quartz (Fig. 6a). Magnesite usually has variable Fe-contents, in parts tracing the former presence of magnetite in the serpentinite mesh, now mostly reacted to Fe-magnesite. The vein network may show a preferred orientation, delineating elongated polygonal mesh cells. In contrast to serpentinite, where euhedral crystal facets of pseudomorphic carbonate are common, in listvenite single veinlets commonly have a corroded appearance or are overgrown by later generations of carbonate Fig. 6a).

Pseudomorphic magnesite-quartz veins after serpentine crack-seal veins ($l_{ss2}$) are characterized by irregular magnesite along vein walls, and columnar to fibrous magnesite extending from the walls into the vein center (Fig. 6b; c.f. Fig. 2h). Magnesite fibers are highly variable in length, ranging from small fractions of the vein aperture to fully bridging the vein. Quartz forms a polycrystalline, non-fibrous aggregate in between the magnesite fibers. This vein microstructure is uncommon for classical antitaxial, syntaxial or stretching veins (Bons et al. 2001), which supports the interpretation that they result from pseudomorphic replacement.

### 4.2.2 Early, zoned magnesite veins

Zoned magnesite veins (lc1, Table 2) are the most abundant in many listvenite core intervals (Fig. 6 c - g) and occur in nearly all studied listvenite samples. As in serpentinites, they form closely spaced parallel or anastomosing branched to crosscutting sets of fibrous to blocky veins. They show a well-defined median zone that is brown in plane-polarized light or contains Fe-oxides or hydroxides and variably strong chemical zoning from the median zone towards the vein walls, indicating antitaxial growth. In most investigated thin sections, lc1 veins are the earliest generation based on cross-cutting relationships (Fig. 6 e - g). In some samples, zoned magnesite veins with a wide-blocky habit form an early generation of this vein type. In some cases, wide-blocky veins are cut by fibrous carbonate veins, whereas in others fibrous segments alternate with wide-blocky sections within a single vein. Owing to these variations in texture that commonly occur together we group them into one vein type (lc1).

In listvenite samples that have a foliated matrix defined by aligned magnesite ellipsoids or dendritic magnesite-quartz intergrowths (Menzel et al., 2021), subparallel clusters of lc1 carbonate veins are oriented at various angles with respect to the matrix foliation. In samples where the vein orientations are at a high angle to the matrix foliation, the veins are locally folded and/or transposed (Menzel et al., 2021).

Lc1 veins are mostly composed of Fe- to Ca-bearing magnesite (Fig. 7). No preserved serpentine inclusions have been observed in this type of veins, but quartz inclusions are common. In some core intervals, where these veins form anastomosing sets, vein segments composed of dolomite alternate with magnesite. The chemical zoning is similar to that of zoned magnesite veins in serpentinite (sc2) and typically bi-symmetric. Lc1 veins commonly have an antitaxial habit, with elongated to fibrous crystals oriented with their long axes perpendicular to the median zone. The median zone usually shows high Fe contents ($X_{Fe}$ = Fe/[Fe + Mg] up to 0.30 in listvenites that contain little or no hematite) near the vein center and becomes progressively Fe-poor

towards the vein walls (Fig. 7b). In some veins, a zone of Ca- and Fe-bearing magnesite ($X_{Fe}$ = 0.10 – 0.15) with rare quartz inclusions occurs along the center of the Fe-rich median zone. In listvenites that contain abundant hematite, Fe-contents in zoned magnesite veins are comparatively low and show less variability, although systematic chemical variations are still apparent owing to variations in minor or trace Ca contents and silica inclusions (Fig. 7f). Similar silica inclusions are common in matrix magnesite of the listvenites, where they have been shown to consist of $SiO_2$ nano-inclusions (Beinlich et al., 2020; Menzel et al., 2021). Fe-enriched domains are commonly brown to red in the core and in plane-polarized light due to Fe-oxides or -hydroxides along the median zone. Cross-cutting relationships show that in some cases the presence of Fe-oxides or –hydroxides is due to oxidation of Fe-magnesite after formation of the lc1 veins (red zones in Fig. 6d; lc1* in Fig. 6g).

SEM-CL images reveal elongated, vein-perpendicular, Fe-rich magnesite in the vein centers (dark-luminescent) and Fe-poor, lighter-luminescent magnesite overgrowths towards the vein walls in crystallographic continuity (Fig. 7d). Many magnesites show euhedral terminations with concentric growth zoning away from the vein center, confirming the antitaxial nature of these veins (Fig. 7d, e). Vein boundaries are irregular with dendritic embayments that extend into the listvenite matrix. A bright-luminescent $SiO_2$ overgrowth rim separates the dendritic embayments from the quartz-rich listvenite matrix (Fig. 7d). This irregular magnesite rim with quartz overgrowth has similar microstructure, luminescence and composition as the outermost, typically dendritic rims around matrix magnesite ellipsoids described by Menzel et al. (2021).

Cross-cutting relationships between different carbonate veins and passive markers in the form of oxides inclusions within the veins show that carbonate vein formation did not only occur by dilatancy but was accompanied by replacement of the serpentinite matrix (Fig. 8 a & b). Measurement of the extent of replacement versus opening is possible in samples where different generations of zoned magnesite veins crosscut each other, because the dilatant vein aperture is recorded by the displacement of the previous vein generation (Fig. 8b). The cumulative vein width is typically more than twice the opening aperture, showing that epitaxial carbonate growth by replacement of serpentine accounts for much of the vein volume. Moreover, Fe-oxides within $Fe^{3+}$-rich listvenite samples and systematic variations in the content of $SiO_2$ may act as passive markers that document vein growth by replacement (Fig. 8 c – e). While hematite may have co-precipitated during serpentine replacement, magnetite (c.f. yellow arrow in Fig. 8d), is a remnant of the prior serpentinization stage and thus a passive marker. Fe-oxides are only cut by dolomite- and quartz-bearing median lines and oxide aggregates are preferentially aligned oblique to the lc1 carbonate veins, recording a previous fabric that appears mostly unaffected by veining. Similarly, many magnetite aggregates were passively overgrown during expansion of carbonate veins. Notably, magnesite ellipsoids in the listvenite matrix have the same, albeit concentric, patterns of silica zoning and similar crosscutting relationships with Fe-oxides (Fig. 8c), indicating that this stage of carbonate growth proceeded similarly and simultaneously in ellipsoidal matrix grains and along vein rims.

**4.2.3 Magnesite and magnesite-dolomite veins**

Magnesite veins (lc2) and magnesite-dolomite veins (lc3) cut zoned lc1 carbonate veins (Table 2; Fig. 6g). They are far less abundant than lc1 veins, and commonly have irregular shapes and boundaries. Lc2 veins are composed of dull or non-luminescent magnesite with comparatively little Fe contents and without chemical zoning. They commonly have cross-fiber to blocky syntaxial textures. Lc3 carbonate veins are composed of polycrystalline magnesite with minor dolomite. Magnesite in lc3 veins may show bright pink luminescence colors under optical-CL, likely due to enrichment in Mn contents, similar to sc3 veins in serpentinite (c.f. Table 1).

**4.2.4 Cryptic quartz veins**

Cryptic quartz veins are one of the earliest quartz generations in listvenite (lq1, Table 2). They are usually indistinguishable from matrix quartz grains in plane- and crossed-polarized light but become visible by CL due to their dull luminescence compared to brighter matrix quartz (Fig. 9 a, b). Cryptic quartz veins commonly have a vermicular, highly irregular and

discontinuous geometry. Many show several stages of growth as revealed by cross-cutting or reactivated zones of different luminescence (Fig. 9b). Locally, CL reveals the presence of a thin, dark-luminescent zone (< 10 μm) with constant thickness over short length scales that could indicate a median zone or refracturing. Notably, most of these veins do not cut zoned magnesite ellipsoids of the listvenite matrix. Instead, they have highly variable thickness and deflect around magnesite
ellipsoids. The cryptic quartz veins usually abut against zoned magnesite veins or exploit their wall – host rock interface. The abundance of this vein type is difficult to estimate due their cryptic nature and because matrix quartz in places shows a similarly dull luminescence, but overall they are less abundant and younger than zoned magnesite veins.

### 4.2.5 Microcrystalline quartz veins

The most enigmatic vein type in listvenites of Hole BT1B are microcrystalline quartz veins (Fig. 9 c – e; lq2, Table 2). They
consist of micro-crystalline, equigranular quartz with a strong crystallographic preferred orientation over long distances, with small variations of the preferred orientation locally producing striped or chess-board patterns under crossed-polarized light. Similar micro-crystalline quartz occurs as variably sized patches in the listvenite matrix suggesting that it may be a replacement microstructure instead of a classic vein infill. SEM-CL imaging shows that the microcrystalline quartz is composed of spheroidal to equant, dull-luminescent quartz grains (3 – 8 μm) surrounded by fibrous, bright-luminescent $SiO_2$ matrix.
Magnesite spheroids are not cut by these veins, but in places occur within them. Inner parts of zoned magnesite veins appear to cut microcrystalline quartz veins and patches. However, botryoidal, euhedral and dendritic carbonate vein rims are undisturbed by the microcrystalline quartz (Fig. 9 d, e), suggesting that at least some of the microcrystalline quartz formed after zoned magnesite veins.

### 4.2.6 Quartz–magnesite and quartz-dolomite veins

Bi-mineralic quartz/chalcedony-magnesite and quartz-dolomite veins (lq4 and lc4 in Table 2) cut earlier quartz and carbonate veins (Fig. 9 c, d). These veins are mostly syntaxial, with sharp, straight vein walls and abundant host-rock inclusions. Wide-blocky carbonate and quartz can be present in the vein center while crystals at the vein walls are smaller, show growth competition textures and commonly have euhedral terminations towards the vein center. Irregular domains of radial chalcedony growth are common, in places also nucleating on the wide-blocky carbonate and quartz crystals along the vein center
(supplementary Fig. S8). Cross-cutting relationships indicate that these veins cut listvenite host rock and are thus younger than carbonation of serpentinite. They are usually older than cataclasites (although in some cases also younger than cataclasis), and are cut by late, open or brecciated carbonate veins (Menzel et al., 2020).

### 4.2.7 Late carbonate veins

The youngest vein generations in BT1 listvenites are mono-mineralic magnesite (lc5) and dolomite (lc6) veins (Table 2). They
have variable microstructures with mostly syntaxial to blocky habits, common euhedral crystal facets and in places open porosity. CL imaging reveals highly oscillatory growth zoning in dolomite within some of these veins (Fig. 9g), which may be due to cyclic variations in Mn incorporation during precipitation, a common phenomenon in calcite veins elsewhere (e.g., Wang and Merino, 1992).

## 5 Discussion

### 5.1 Sequence of reactions and vein formation

Vein microstructures in carbonated peridotites are key to understanding the coupled feedbacks between deformation, fluid flow, and carbonation, and may provide valuable insights for industrial carbon storage by mineral carbonation (van Noort et al., 2013). In the BT1B listvenites and serpentinites, vein-fill minerals may have formed due to (i) precipitation from

supersaturated fluids along fluid pathways in serpentinite during an incipient stage of carbonation; (ii) precipitation of the reaction products magnesite and/or quartz *during* in-situ dissolution and replacement of the host serpentine; and (iii) precipitation along fractures in listvenite after termination of the actual carbonation reaction. Microstructural evidence and cross-cutting relationships presented in this study indicate that pseudomorphic veins, zoned magnesite veins and cryptic and microcrystalline quartz veins are coeval with different stages of the carbonation reaction sequence that consumes serpentine (Fig. 10). Some of the textures and cross-cutting relationships are ambiguous, but in general terms a first stage of carbonate veining preceded extensive crystallization of quartz in veins and the listvenite matrix. Therefore, syn-carbonation veining must have occurred in the early upper Cretaceous coinciding with the main timing of listvenite formation (Falk & Kelemen, 2015). We distinguish the following stages of microstructural evolution:

I   Early, high temperature (T > 700 °C) deformation of the banded peridotite protolith, producing a fabric with elongated and aligned Cr-spinel and orthopyroxene. The protolith was partly refertilized through high-T metasomatism (Godard et al., 2021) that is typical of the basal peridotites in the Samail ophiolite (Prigent et al., 2018).

II  Serpentinization of olivine and pyroxene to form mesh and bastite serpentine, respectively, likely at T < 250 °C with formation of magnetite and, in dunitic protolith compositions, brucite. Deformation after and possibly also during serpentinization caused ductile shear zones with aligned lizardite and flattened magnetite mesh structures (Menzel et al., 2021). In places, serpentinization may have been accompanied by cataclasis, similar to that in partially serpentinized peridotite of the Wadi Tayin massif (Aupart et al., 2021).

III Formation of (not mesh) serpentine-magnetite and banded serpentine crack-seal veins (ss1, ss2; Table 1). Cleavage-parallel serpentine veins formed in foliated serpentinites (Fig. 2d), although they may be obscured by carbonate in BT1B.

IV  Incipient precipitation of carbonate, in particular Fe-rich magnesite, as ellipsoidal/spheroidal grains in the serpentine matrix (Beinlich et al., 2020b), along the outlines of polygonal mesh cells (sc0 in Fig. 2, Fig. 10) and in early carbonate veins (sc1 and the median zone of sc2 veins; Table 1). The ellipsoidal/spheroidal grain habit is interpreted to be a result of disequilibrium precipitation at high oversaturation (Beinlich et al., 2020b) and/or under deviatoric stress (Menzel et al., 2021). Remnant olivine, pyroxene and brucite after serpentinization may have reacted preferentially with $CO_2$ to form carbonate.

V   Locally (about 10 – 15 % of core BT1B): ductile deformation of the reacting, serpentine-bearing assemblage, leading to folding of early carbonate veins and development of a penetrative foliation by oriented growth of ellipsoidal magnesite in the matrix (Menzel et al., 2021).

VI  Concentric growth of matrix magnesite grains and widening of zoned magnesite veins by replacement of the serpentine matrix and/or opening, in places with precipitation of some talc or quartz (Fig. 4). This is consistent with the microstructures of overgrowths on folded magnesite veins in ductilely deformed listvenites from Hole BT1B, which indicate carbonate vein opening and deformation occurred before listvenite formation was completed (Menzel et al., 2021). This stage may have been accompanied by silica loss on a local scale.

VII Incipient precipitation of quartz in the remaining serpentine matrix and formation of early, syn-carbonation quartz veins (sq1 in Table 1; lq1 and lq2 in Table 2). In places, opal may have precipitated initially and later recrystallized to quartz or chalcedony (Kelemen et al., 2022).

VIII Dendritic growth of magnesite on ellipsoidal matrix grains and along the walls of early carbonate veins (Fig. 7; Fig. 9 e) and precipitation of cryptic and/or microcrystalline quartz in veins and the matrix. Complete replacement of remnant matrix serpentine by quartz and minor carbonate concluded the carbonation reaction.

IX      Syntaxial to blocky quartz–carbonate veins that cut listvenite and, rarely, serpentinite (lq4, lc4; Fig. 9 f, g; Fig. 10).

Quartz proportions in these veins are typically higher than carbonate, pointing to silica influx or redistribution during this stage. It is possible that the formation of bimineralic quartz-carbonate veins (lq4, lc4) occurred in listvenite while carbonation proceeded at the advancing reaction front along the serpentinite-listvenite contact. Alternatively, they may have formed due to fracturing during a first deformational overprint following carbonation.

X       Cataclasis, sharp faults and late carbonate veins overprinting listvenite and serpentinite (Fig. 10; sc4, lc5, lc6 in Table

1 & 2), in parts related to local Ca gain and Mg loss in listvenite (Menzel et al., 2020).

In terms of chemical reactions, we infer that the main stages of progressive transformation to listvenite proceeded by the following simplified reactions:

(1) Hydration of olivine and orthopyroxene to serpentine and brucite, forming serpentine mesh veins and bastite (stage II), with brucite proportions depending on orthopyroxene abundance (for $0 \leq m \leq 1$):

$$(2 - m)\, Mg_2SiO_4 + m\, MgSiO_3 + (3 - m)\, H_2O = Mg_3Si_2(OH)_4 + (1 - m)\, Mg(OH)_2 \tag{R1}.$$

(2) Incipient magnesite formation in the matrix, in sc0 veins and the center of sc1 and sc2 veins (stage IV – VI), by reaction of brucite and serpentine with $CO_2$; initially likely with little formation of secondary silicates but related to transfer of aqueous silica ($SiO_{2,aq}$; for $0 \leq n \leq 2$):

$$Mg(OH)_2 + CO_{2\,(aq)} = MgCO_3 + H_2O \tag{R2},$$

$$Mg_3Si_2(OH)_4 + 3\, CO_{2\,(aq)} = 3\, MgCO_3 + (2 - n)\, SiO_2 + n\, SiO_{2\,(aq)} + 2\, H_2O \tag{R3}.$$

Isochemical replacement of serpentine by magnesite and quartz (n = 0 in R3) would lead to a magnesite/quartz proportion of 1.5 molar, equivalent to ~34 vol% quartz. Local mobility of aqueous silica is inferred from the observation that many matrix domains and magnesite veins have magnesite/quartz proportions significantly higher than 1.5 molar (Fig. 3; Fig. 4). Thus, only some of the released silica precipitated in-situ, forming $SiO_2$ inclusions in magnesite or, rarely, talc (e.g., Fig. 4), indicating

local leaching of $SiO_{2,aq}$. In addition or alternatively to silica loss, reaction (3) may be complemented by influx of aqueous Mg-bearing species, and local Ca mobility where early dolomite is present (c.f. Beinlich et al., 2020b).

(3) Reaction of serpentine with $CO_2$ to form magnesite and quartz (stages VII – VIII), locally with excess influx of aqueous silica derived from reaction R3 forming syn-carbonation quartz veins and (microcrystalline) quartz-rich domains:

$$Mg_3Si_2(OH)_4 + 3\, CO_{2\,(aq)} + n\, SiO_{2\,(aq)} = 3\, MgCO_3 + (2 + n)\, SiO_2 + 2\, H_2O \tag{R4}.$$

(4) Post-listvenite syntaxial and late veins in listvenite (stages IX – X) do not show replacement structures involving reaction of serpentine with $CO_2$, but are inferred to have formed primarily by precipitation from aqueous solutions with variable concentration of dissolved Si, Ca, Mg and bicarbonate/carbonate:

$$Mg^{2+} \pm Ca^{2+} \pm HCO_3^- \pm CO_3^{2-} \pm SiO_{2\,(aq)} = MgCO_3 \pm SiO_2 \pm (Ca,Mg)CO_3 \pm CaCO_3 \tag{R5}.$$

Some of the dolomite locally replacing early magnesite veins and filling late veins may be reflective of magnesite dissolution

in combination with influx of Ca-bearing fluids, possibly related to fluid flow during brittle overprint (Menzel et al., 2020). Some of the youngest, post-listvenite magnesite and dolomite vein generations (lc5, lc6) may have a similar origin to very young magnesite, dolomite and calcite/aragonite veins related to interaction of Mg-$HCO_3^-$ and Ca-$OH^-$ bearing groundwater with the Samail peridotite (Noël et al., 2018; Streit et al., 2012; Ternieten et al., 2021).

Because the reaction of serpentine to magnesite and quartz (reactions R3+R4) consumes $CO_2$ while releasing $H_2O$, the fluid

evolves to more aqueous compositions with reaction progress. Thus, steps II – VIII and reactions R1 – R4 may have occurred at the same time along different advancing reaction fronts, which correspond to the contacts between partially hydrated peridotite, serpentinite, carbonate-bearing serpentinite and listvenite.

We note that not every stage I – X is recognizable in each core section. Moreover, field exposures around site BT1 show high variability of strongly veined domains alternating with more massive listvenite intervals. We further note that vein

microstructures in dolomite- and dolomite-calcite listvenites that are common further north in the Fanjah region are somewhat

different from the magnesite-dominated BT1B listvenites studied here, as they are related to a massive gain of Ca through fluid influx (reaction R5).

## 5.2 Influence of pre-existing serpentine structures on veining

Pseudomorphic carbonate after mesh and crack-seal serpentine veins (Fig. 2 e – h; Fig. 6 a, b) demonstrate that the
microstructure of the precursor serpentinite determined the location and structure of vein networks to a great extent. The local presence of brucite and/or variations in Si, Al and Fe contents of serpentine may have caused preferential carbonation at specific microstructural sites where carbonation reaction affinity is higher. Brucite, which shows very fast carbonation reaction kinetics in low-temperature experiments (Harrison et al., 2013; Hövelmann et al., 2012), is commonly observed together with magnetite along serpentine mesh veins in serpentinites (Schwarzenbach et al., 2016). The preferential replacement of previous
brucite by magnesite (reaction R1) may thus explain the polygonal, mesh-pseudomorphic carbonate vein network in some serpentinites and listvenites of Hole BT1B (Fig. 2 e – f; Fig. 6a). However, brucite is typically only abundant in serpentinized dunite, because its stability requires high Mg/Si of the bulk rock. As large parts of the listvenites of Hole BT1B are inferred to have had a serpentinized harzburgite and lherzolite protolith, based on major and trace element geochemistry (Godard et al., 2021), we infer that brucite was only common in minor dunitic intervals. On the other hand, different parts of mesh
microstructures and different veins in serpentinite can be composed of a variety of serpentine polytypes with different crystal structure. Acid-leaching experiments have shown that dissolution rates can differ greatly between these polytypes, with much higher Mg extraction rates for chrysotile, nano-tubular chrysotile and poorly ordered lizardite compared to Al-bearing lizardite, polygonal serpentine, and antigorite (Lacinska et al., 2016). It is therefore likely that different serpentine polytypes also show variable dissolution rates during reaction with moderately acidic, $CO_2$-bearing aqueous fluids. This may explain why specific
microstructural sites are preferentially replaced by carbonate, producing pseudomorphic textures. Besides variable dissolution rates, serpentine polytypes also have different crystal habits with differing strength and surface area. Thus, we propose that the heterogeneous microstructures of different serpentine polytypes form micro-environments with different inter- and intra-granular nano-porous matrix permeability and micron-scale permeability along fractures. These heterogeneities create complex relationships between diffusive and advective solute transfer, fluid-flow rates and kinetics that control different levels of
pseudomorphic inheritance.

Banded serpentine crack-seal veins appear to have a particularly strong impact on local fracture formation and small-scale porosity morphology. Such serpentine veins typically consist of chrysotile fibers alternating with lizardite or polygonal serpentine, recording repeated crack-seal cycles (Andreani et al., 2004; Tarling et al., 2021). Due to the high tensile strength of chrysotile parallel to fiber orientations, fracturing and associated permeability is expected to occur preferentially along the
vein – host rock interface, which is what we observed in pseudomorphic replacement microstructures (Fig. 2h; Fig. 6b).

## 5.3 Vein growth mechanisms — opening versus replacement

Opening of dilatant fractures can increase permeability and provide pathways for fluid infiltration that would allow carbonation to proceed. What type and how much permeability is created, however, depends on how soon after opening and in which direction the vein becomes filled. When crystals precipitate at the vein walls and grow inwards towards the vein center
(syntaxial veins), fluid replenishment and, potentially, crack-seal events occur along the vein center (Bons et al., 2012). This potentially results in a loss of connectivity between the vein and matrix permeability network because of mineralization along the vein-matrix interface. In contrast, if crystal growth proceeds from a median zone towards the vein walls (antitaxial veins), fluid flow is focused along the vein-host rock interface creating a connected permeability network between the fracture and rock matrix. We found examples of both types of vein growth in the BT1B serpentinites and listvenites (Tables 1 & 2). In
general terms, early serpentine and carbonate veins (e.g., Fig. 2, Fig. 4) as well as some early quartz veins (Fig. 5b) tend to show antitaxial textures, whereas younger quartz-carbonate veins (lq4, lc4; Table 2) tend to be syntaxial. If the process was

entirely mechanical, this would suggest a reduction in the connectivity of the permeability network over time. Owing to chemical-mineralogical replacements that occur during carbonation, however, the mechanism is more complex during listvenite formation.

Current models of vein formation treat the host rock as a non-reactive substrate with vein formation due to precipitation from aqueous solution in fluid-filled fractures (Ankit et al., 2015; Hilgers et al., 2001; Hubert et al., 2009; Spruženiece et al., 2021a; Spruženiece et al., 2021b). This mode of vein formation is commonly observed in carbonate veins in altered oceanic crust (e.g., Quandt et al., 2020). In the case of carbonate veining during listvenite formation, however, mechanical opening was accompanied by replacement of the host serpentinite (Fig. 8) so that the morphology of the fracture wall is controlled by

dissolution and replacement in addition to dilatancy. Therefore, the wall rock changes its morphology by dissolution, and vein volume is further accommodated by replacement. In addition to evidence from cross-cutting relationships and overgrown passive markers within veins and the listvenite matrix (Fig. 8), the high abundance of $SiO_2$ nano-inclusions within zoned magnesite veins in listvenite of Hole BT1 as confirmed by transmission electron microscope (TEM) (Beinlich et al., 2020b; Menzel et al., 2021) indicates that silica saturation and quartz nucleation rate were high during carbonate vein growth, which

provides further evidence for simultaneous serpentine dissolution.

Microstructures indicative of growth zoning during carbonate vein growth from a median zone outward into the serpentine matrix (Fig. 7d) suggest that there was a reactive fluid film and significant permeability along the vein – host rock interface (Fig. 11). Compared to the fracture permeability created initially by dilatant opening of the vein, which may easily clog if mineral precipitation is fast, we postulate that this interface permeability was maintained by vein growth by replacement and

coupled dissolution of serpentine. Facetted carbonate crystal terminations, partial talc infills and secondary exploitation by quartz veins (Fig. 4) suggest that the vein-serpentinite interface was a preferential site of focused, advective fluid flow and, in places, new fracture formation. This interface permeability thus promoted continued vein growth by serpentine dissolution, in addition to supplying $CO_2$-bearing fluid to the nano-porous matrix of the non-veined host serpentine through diffusive solute transfer, facilitating progressive carbonation of serpentinite. Subsequent syntaxial quartz-carbonate veins most likely lacked

such a reaction front, with fluid pathways concentrated along the center of the vein.

### 5.4 Formation of closely spaced carbonate vein sets

Because early, zoned magnesite veins are extremely abundant in serpentinite and listvenite of Hole BT1B, and because of their likely role in acting as main fluid pathways early in the carbonation process, understanding their formation mechanism is integral to deciphering the factors controlling carbonation reaction progress. A key feature of these carbonate veins is that they

commonly form closely spaced, subparallel sets. Similar, subparallel serpentine vein sets occur in serpentinites in the vicinity of listvenites in the area (Fig. 2b) and are common in other serpentinized peridotites of the Samail ophiolite (Kelemen et al., 2020a, c). Parallel, closely spaced serpentine vein sets are also known from oceanic peridotites (Andreani et al., 2007). Repeated fracturing parallel to existing veins requires that the veins and the vein-host rock interface are stronger than the host rock (Virgo et al., 2014). However, the reaction front at the vein – serpentinite interface of zoned magnesite veins (sc2 and lc1

veins; Fig. 11) speaks against a strong vein – host rock interface during this stage of reaction. Furthermore, the zoning patterns, documenting changing fluid compositions and/or redox conditions, are consistent within different veins of the same set. Hence, a sequential process of repeated parallel fracturing and sealing by zoned carbonate growth is unlikely, because it would require similar, cyclic variations of fluid composition and redox conditions to be repeated for each vein.

A more feasible explanation is that the zoned parts of the carbonate veins formed along a preexisting fracture or vein set. If

the vein material had a higher strength than the host serpentinite, closely spaced vein sets may form. This may have happened if the initial vein fill had a higher permeability or higher carbonation reaction affinity than serpentine of the host rock, so that the veins preferentially became replaced by carbonate. Zoned carbonate growth may then have proceeded from the narrow vein set into the serpentine matrix in a later step. A precursor vein fill of fibrous chrysotile may be a suitable candidate because

chrysotile has the same or higher tensile strength compared to matrix serpentine, facilitating fracturing parallel to existing veins. Chrysotile veins may also show higher carbonation reaction rates than lizardite due to the larger surface area of fibrous aggregates, especially if they have a nano-tubular crystal morphology (Lacinska et al., 2016), in line with the observation of other pseudomorphic carbonate veins (Fig. 2 c, h). High-resolution SEM imaging of a broad-ion-beam polished sample of veined serpentinite (see methods) reveals that fibrous serpentine veins can have substantially higher micro- to nano- porosity than the matrix serpentine (Fig. 12), confirming that they may be sites of preferential reactive fluid flux. Similar subparallel serpentine vein sets may thus have been the precursor of the closely spaced sc2 and lc1 carbonate veins. Such serpentine veins may form by repeated fracturing of serpentinite, or during serpentinization of olivine when tectonic stress enhances widening of favorable orientations of mesh veins (c.f. Fig. 5 of Aupart et al. (2021)).

## 5.5 Reaction-induced fracturing?

Listvenites are inferred to form, among other settings, at the base of ophiolites or in the shallow mantle wedge of subduction zones (e.g. Kelemen & Manning, 2015). At these conditions, all principal stresses will be compressive. Thus, fracturing by tensile or shear failure typically requires a reduction of effective stress by fluid overpressure (Hilgers et al., 2006; Sibson, 2017). Experiments and numerical models of volume-expanding hydration reactions have shown that crystallization pressure may locally create gradients in differential stress, which can also facilitate fracture formation, increasing permeability and reactive surface area (Malthe-Sørenssen et al., 2006; Rudge et al., 2010; Shimizu and Okamoto, 2016). In combination with elevated fluid pressure, these local stress gradients caused by "force of crystallization" could lead to dilatant opening and propagation of existing veins, formation of new fractures, or enhanced pressure solution of the rock matrix (Fletcher and Merino, 2001). Kelemen and Hirth (2012) propose that crystallization pressure during peridotite carbonation can be large enough to exceed the stress required for frictional failure, creating a positive feedback for reaction progress via fracturing. This process has been shown to be efficient for reactions where volume changes are very large, such as during hydration of periclase (MgO) to brucite (Zheng et al., 2019; Zheng et al., 2018) and important during serpentinization of olivine (Evans et al., 2020; Plümper et al., 2012; Yoshida et al., 2020). However, the extent to which crystallization pressure influences listvenite formation is less certain (van Noort et al., 2017). Full hydration and carbonation of olivine increases the solid volume by 33% and > 40% (Kelemen et al., 2011), respectively, while the conversion of serpentine to magnesite and quartz is predicted to cause a solid volume expansion of 18 – 22 % (Hansen et al., 2005; Kelemen et al., 2022). Reaction-induced fracturing due to crystallization pressure and volume expansion may thus have occurred at the advancing serpentinization and carbonation reaction fronts that formed the serpentinites and listvenites at site BT1.

Zoned magnesite veins may theoretically have opened through crystallization pressure to some extent, because, unlike in syntaxial veins, carbonate growth occurred from the center outwards. However, several observations argue against this mechanism dominating during early carbonation: (i) passive markers show that much of the vein width was accommodated by replacement rather than opening (Fig. 8), (ii) euhedral growth patterns point to the presence of an open fluid conduit at the vein-matrix interface (Fig. 11), and (iii) most zoned magnesite veins in serpentinites and listvenites contain a much smaller proportion of $SiO_2$ (mostly as inclusions in magnesite) than expected for isochemical replacement of serpentine (Fig. 4, Fig. 7, Fig. 8), indicating that silica was leached (reaction R3). Mg isotope geochemistry and bulk chemistry mass balance calculations suggest that Mg in listvenite magnesite is derived from local dissolution of the peridotite protolith (de Obeso et al., 2021; Godard et al., 2021). Assuming that external Mg influx was negligible and that magnesite growth is rate-limited by serpentine dissolution, reaction R3 would cause volume expansion only if at least ~22 vol% of the solid reaction products is quartz. If more $SiO_{2,aq}$ is leached than precipitated in-situ as quartz (n > 1 in reaction R3), solid volume change would be negative. Since in-situ quartz abundance in zoned magnesite veins is typically < 10 vol%, combined influx of $CO_2$ and local leaching of silica probably resulted in a solid volume decrease at the vein-serpentine interface because magnesite has a higher density than serpentine. The leached silica may have precipitated synchronously in different microstructural sites in the rock

matrix, forming quartz-rich domains, or as cryptic, microcrystalline or syntaxial quartz veins (Fig. 8; reaction R4) further downstream along the reaction front or in listvenite. Abundant $SiO_2$ nano-inclusions in magnesite point to widespread quartz oversaturation and high nucleation rates. This suggests a non-trivial coupling between the surface properties, porosity and dissolution rate of serpentine and the interface geometry, solute transport and precipitation kinetics during vein growth.

Possibly, some local silica mobilization occurred in the form of suspended silica nano-aggregates at high fluid-flow rates. The occurrence of centimeter-scale bulk chemical variations in the BT1B listvenites suggests that similar local mass transfer was commonplace during listvenite formation (Godard et al., 2021).

Numerical models suggest that volume-increasing reactions with fast reaction kinetics induce polygonal and hierarchical fracture patterns (e.g., Okamoto and Shimizu, 2015; Ulven et al., 2014), in agreement with the typical mesh textures in

serpentinites. In contrast, the BT1B zoned magnesite vein sets have parallel or anastomosing patterns that indicate a strong influence of tectonic stress during initial fracture formation.

Taken together, these observations suggest that although volume expansion associated with the overall transformation to listvenite (combined reactions R3 + R4) caused some crystallization pressure, zoned magnesite veins did not primarily grow through force of crystallization. Crystallization pressure may have contributed to the external stress responsible for the initial,

dilatant fractures along which the carbonate veins developed. A similar conclusion can be drawn from the microstructures of pseudomorphic carbonate (sc0) and feathery, cleavage-parallel carbonate veins (sc1) in carbonate-bearing serpentinite (Table 1): quartz is rare or absent in their vicinity, indicating that their formation did not require volume expansion if Mg was sourced locally from dissolving serpentine. Reaction-induced fracturing was however likely prevalent during the preceding, highly volume-expanding serpentinization, which created mesh and vein textures with heterogeneous permeability and carbonation

affinity.

## 5.6 Tectonic context

$CO_2$ fluid flux derived from subduction/underthrusting of (meta)sediments below the Samail ophiolite is considered the most likely setting of listvenite formation at Site BT1 (de Obeso et al., 2022; Kelemen et al., 2022), although carbonation during extensional reactivation of the thrust fault in an early phase after obduction is possible (c.f. section 2.2). While the common

parallelism of zoned magnesite veins points to a strong influence of tectonic stress on vein formation, our results do not allow us to determine whether veining occurred in an overall contractional or extensional setting. This caveat is due to unoriented drill cores and the complexity arising from the observation that most of the syn-carbonation veins are replacement and not purely dilatant veins. Folding (Kelemen et al., 2022) and several phases of post-listvenite brittle faulting (Menzel et al., 2020) further complicate a reconstruction of paleo-stress directions during formation of the different vein generations. Based on

ductile deformation microstructures, the morphology of preserved micro- and nano-pores and the requirement for high fluid fluxes to form listvenite, Menzel et al. (2021) inferred that pore pressures were high during carbonation. High pore pressures can lead to tensile and extensional fracturing also in thrust settings, which may be particularly widespread at contacts and in mixtures of lithologies with disparate rheology (e.g. Sibson, 2017). Such conditions may be common at the reaction front along the contacts of comparatively strong listvenite lenses enclosed in weak, sheared serpentinite. In Oman, fluid overpressure cells

formed in response to burial below the ophiolite and the Hawasina nappes, causing veining in (par)autochthonous carbonate rocks (Grobe et al., 2019). The widespread and multi-phase syn-carbonation veining observed in the BT1B serpentinites and carbonates may thus be a combined effect of reaction-induced stress from serpentinization and cyclic fluctuations between sub-lithostatic and lithostatic fluid pressures in the deforming and reacting basal peridotites during subduction and/or ophiolite emplacement. With subsequent progressive cooling of the ophiolite and underlying units, $CO_2$ concentration in infiltrating

fluids likely decreased. Thus, the fluid chemistry may have switched to more Ca- and/or Mg- bicarbonate ionic solutions favoring the formation of syntaxial, post-listvenite veins and precipitation of carbonate vein fill, in contrast to the earlier syn-listvenite replacement veins that formed by reaction of serpentine with $CO_2$.

**6 Conclusions**

Microstructures and cross-cutting relationships in serpentinites and listvenites of Hole BT1B demonstrate that several vein generations formed during carbonation reactions. These veins constitute large volumes of the BT1B listvenites, showing that fracturing and related advective fluid flow were integral to carbonation progress. The incipient stages of carbonation are consistently related to pseudomorphic carbonate and zoned magnesite veins. Zoning of Ca, Fe and Si contents in early carbonate veins records variations in fluid composition, changes in redox conditions, and variations of supersaturation and nucleation rates of silica during progressive serpentine replacement. Cross-cutting relations and passive markers indicate that zoned magnesite veins formed as incipient dilatant, often parallel and closely spaced micro-fractures, possibly initially filled with precursor chrysotile that was preferentially replaced by carbonate. From this incipient median zone, a permeable micro-reaction front developed into the serpentine matrix upon further $CO_2$ influx allowing vein growth to continue through a dilatant-reactive process. These observations indicate that vein – wall rock interfaces served as essential fluid conduits during transformation of the non-veined matrix into listvenite. Sets of parallel to anastomosing carbonate veins point to an important role of tectonic stress during early carbonation, likely complemented by deviatoric stress generated by volume expansion at the serpentinization front advancing ahead of the carbonation reaction front, whereas crystallization pressure from magnesite precipitation was most likely less significant during veining. As carbonation progressed, permeability was probably reduced during subsequent quartz veining and further silica replacement of the matrix, but a lack of remnant serpentinite in listvenite horizons indicates that penetration of $CO_2$-rich fluid through the vein and matrix permeability network was sufficient for carbonation to proceed to completion. Our results suggest that in this natural example, veining caused by tectonic stress and fluid overpressure is an important mechanism to create permeability despite carbonate precipitation. Without the added effect of tectonic deformation and related deviatoric stress, it is possible that permeability created through reaction-driven fracturing +/- replacement veining alone is not enough to allow for the necessary fluid-flux for carbonation to progress. Therefore, the extent of carbon mineralization and permanent $CO_2$ sequestration that can be attained via experimental in-situ $CO_2$ injection might be limited.

**Sample and data availability**

Archive halves and samples of core BT1B are available through the Oman Drilling Project (https://www.omandrilling.ac.uk/samples-data). Digitized thin sections of some of the samples used in this study are available for download as Supplementary material of the Oman Drilling Project (http://publications.iodp.org/other/Oman/SUPP_MAT/index.html#SUPP_MAT_Z). All main data is contained in the figures and tables of the manuscript and the supplementary material; raw images of figures are available from the authors upon request.

**Author contribution**

MDM and JLU designed the study, conducted field work and studied the microstructures and petrography; MDM and TD refined the vein classification; MDM performed SEM imaging, EDX mapping, optical CL analysis and image processing, and drafted the figures; EU conducted SEM and SEM-CL analysis. All authors discussed and interpreted the results. MDM led writing and revision of the manuscript, to which all authors contributed.

**Competing interests**

The authors declare that they have no conflict of interest.

**Acknowledgments**

We would like to thank Michael Kettermann and Yumiko Harigane for sampling onboard Chikyu, and Peter Kelemen for providing an invaluable set of additional thin sections. Werner Kraus and Jonatan Schmidt are thanked for thin section preparation and technical assistance, and Sara Elliott for assistance with SEM-CL imaging and post-processing. We are grateful to the Oman Public Authority of Mining for support to conduct field work and sample export. This study has benefitted from fruitful and inspiring discussions with Peter Kelemen, Romain Lafay, Juan Carlos de Obeso, Craig Manning and others over the past years. We further acknowledge the constructive reviews by Dennis Quandt, an anonymous referee and topical editor Virginia Toy, whose comments helped to improve several aspects of the paper.

MDM and JLU acknowledge funding by the German Research Foundation (DFG grants UR 64/20-1, UR 64/17-1). This research used samples and data provided by the Oman Drilling Project. The Oman Drilling Project (OmanDP) has been possible through co-mingled funds from the International Continental Scientific Drilling Project (ICDP), the Sloan Foundation – Deep Carbon Observatory (Grant 2014-3-01), the US National Science Foundation (NSF-EAR-1516300), NASA – Astrobiology Institute NNA15BB02A), the German Research Foundation (DFG: KO 1723/21-1), the Japanese Society for the Promotion of Science (JSPS no:16H06347; and KAKENHI 16H02742), the European Research Council (Adv: no.669972), the Swiss National Science Foundation (SNF:20FI21_163073), the Japanese Marine Science and Technology Center (JAMSTEC), the International Ocean Discovery Program (IODP), and contributions from the Sultanate of Oman Ministry of Regional Municipalities and Water Resources, the Oman Public Authority of Mining, Sultan Qaboos University, CRNS-Univ. Montpellier II, Columbia University of New York, and the University of Southampton.

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

**Figures**

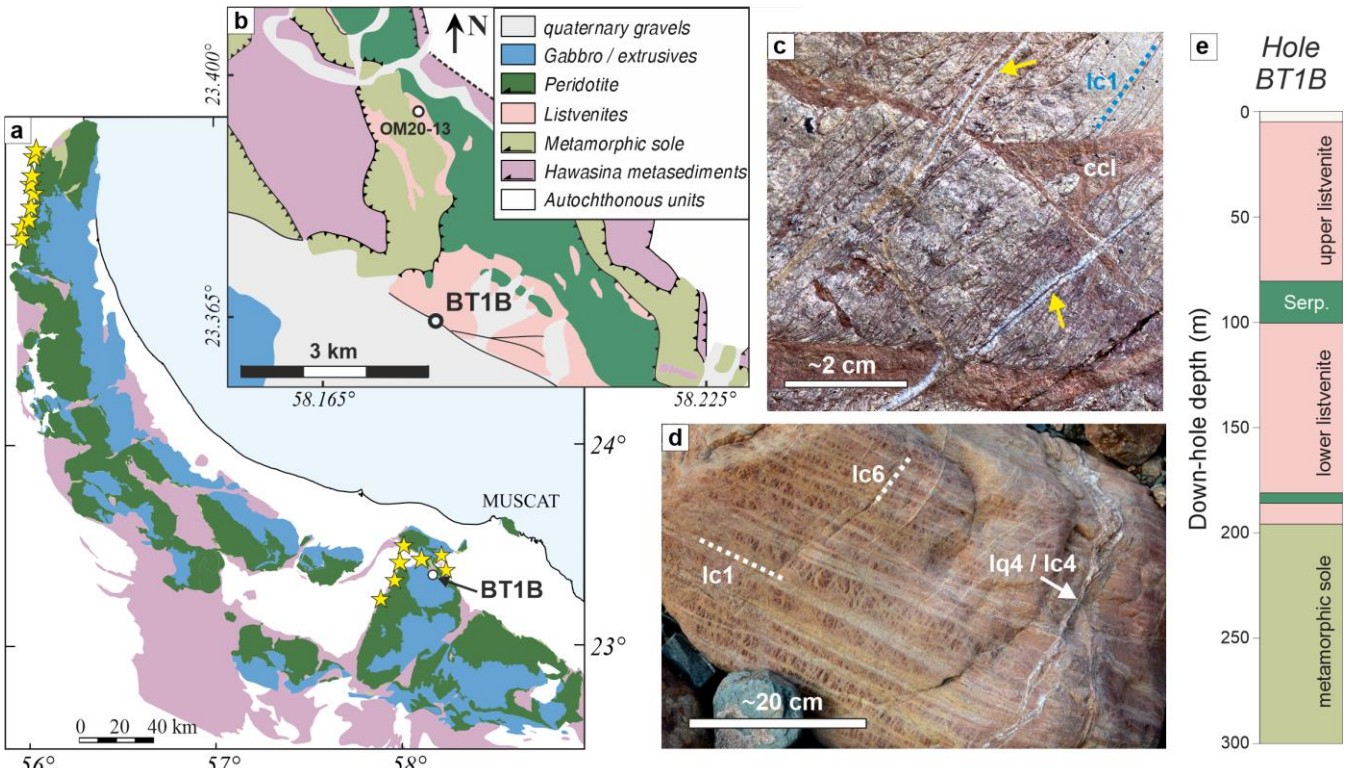

**Figure 1: Geological overview map of the Samail ophiolite (common legend in b), with locations of known listvenite occurrences (yellow stars) (after Nicolas & Boudier, 1995). (b) detailed geological overview of the area surrounding Site BT1 (after de Obeso et al., 2022 and Villey et al., 1986); (c) Vein generations in outcrop, with closely-spaced narrow carbonate veins (lc1, blue dotted line) cut by cataclasite (ccl, brown/red) and quartz-carbonate veins (yellow arrows). (d) Different vein generations in listvenite boulder with exceptionally wide veins, with vein network in the matrix resembling a mesh cut by closely spaced parallel carbonate (lc1) and later quartz-carbonate (lq4 / lc4) and dolomite veins (lc6). (e) Lithologies in Hole BT1B (Kelemen et al., 2020b).**

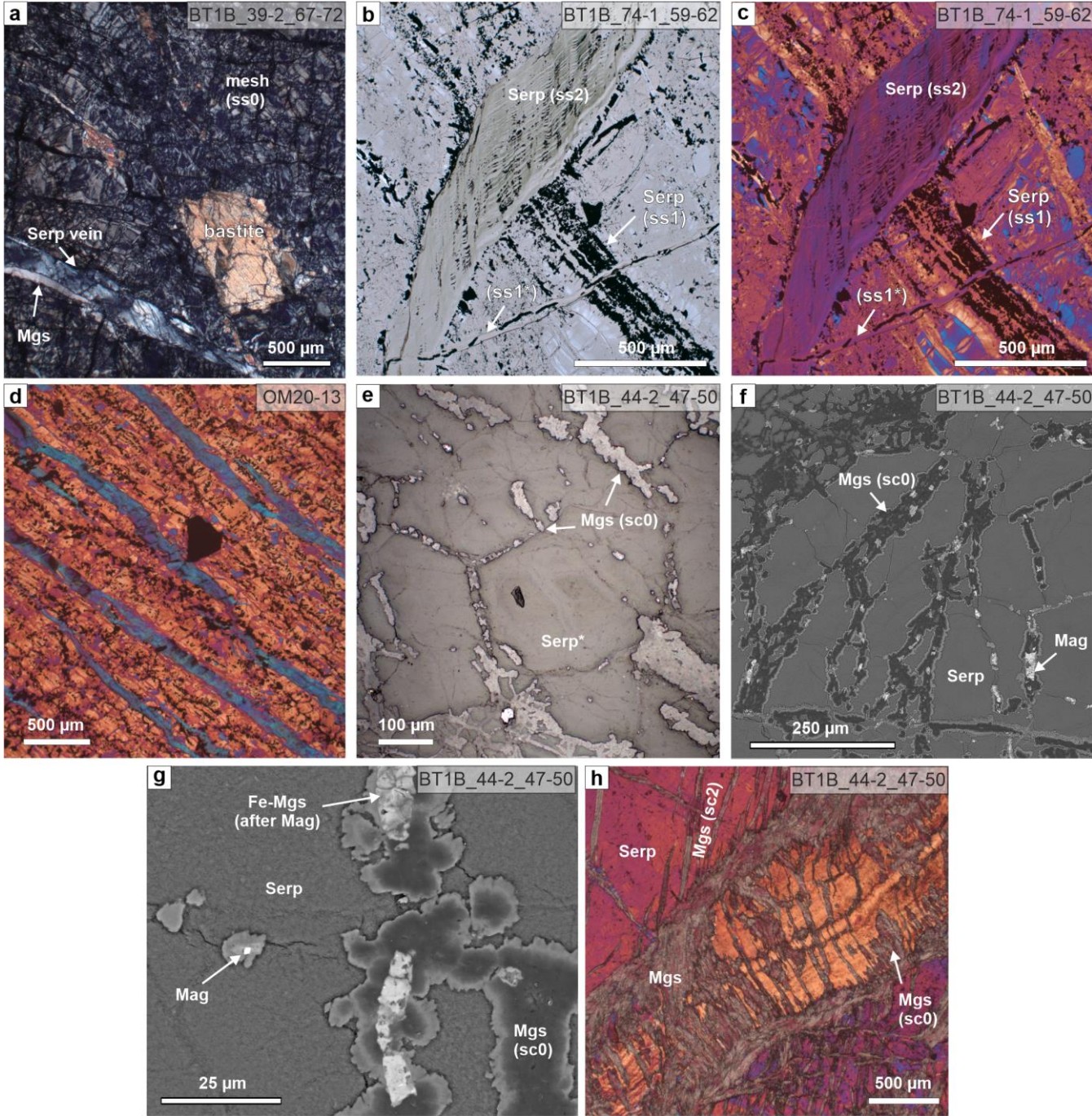

**Figure 2: Serpentine veins (a - d), and pseudomorphic carbonate microstructures after serpentine textures (e & f) in serpentinite.**
**(a) Mesh (ss0) and bastite textured serpentinite, cut by serpentine vein; a magnesite vein exploits the vein-host rock interface of the serpentine vein under crossed polarizers (xpol). (b) Plane-polarized (ppol) micrograph of wide serpentine-magnetite vein (ss1) cut by a light-green, banded serpentine crack-seal vein (ss2), which is in turn cut by a narrow serpentine-magnetite vein (ss1*). (c) Same area as in b, with xpol and 1λ-plate. (d) Clustered cleavage-parallel serpentine veins in foliated serpentinite north of site BT1 with strong crystallographic preferred orientation and flattened magnetite mesh cells. The large black grain is Cr-spinel. (xpol with 1λ-plate). (e) Reflected light image and (f) BSE image of magnesite vein network (sc0) tracing polygonal serpentine mesh (Serp*) delineated by former magnetite in serpentinite. (g) Detailed BSE image of sc0 magnesite showing pseudomorphic replacement of magnetite by Fe-magnesite (similar replacements are known from listvenites elsewhere; Menzel et al., 2018), and compositional zoning with Fe-bearing euhedral to dendritic magnesite rims. (h) Serpentine crack-seal vein (orange) in serpentinite, with partial pseudomorphic replacement by magnesite and crosscut by zoned magnesite-dolomite veins (sc2) (xpol with 1λ-plate). For abbreviations of minerals and vein generations used in all figures, see Tables 1 & 2.**

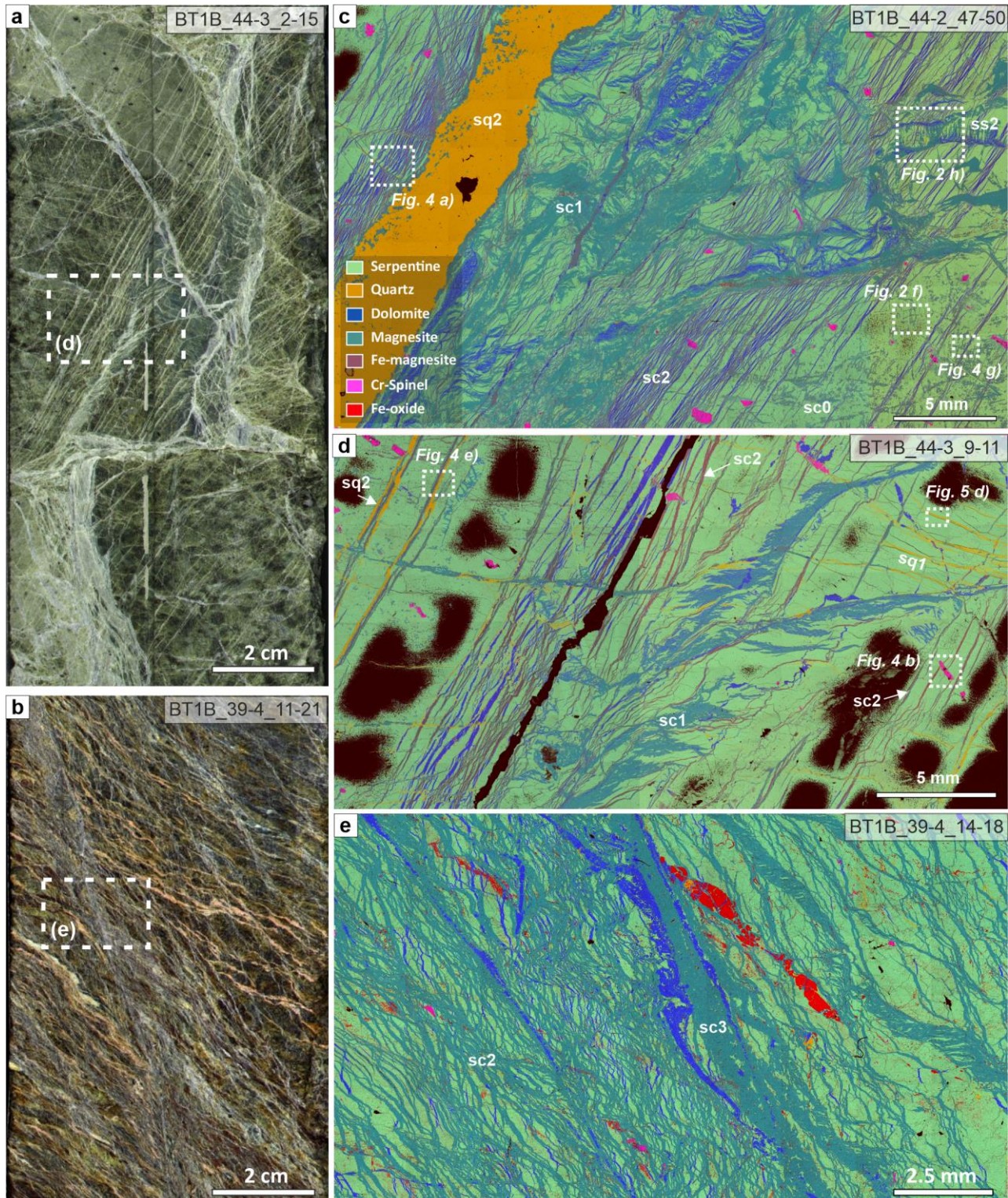

**Figure 3: Veins in serpentinite of Hole BT1B. (a)** Split-core image of serpentinite with closely spaced, parallel carbonate veins. **(b)** Split-core image of strongly veined serpentinite interval with anastomosing carbonate veins. **(c - e)** Composite-color EDX maps of carbonate-rich serpentinites showing different carbonate (sc1 - sc3) and quartz (sq1, sq2) vein generations (common legend in c). Black patches in d) are areas where soft bastite serpentine were lost during sample preparation. Core diameter in a) and b) is 6.3 cm.

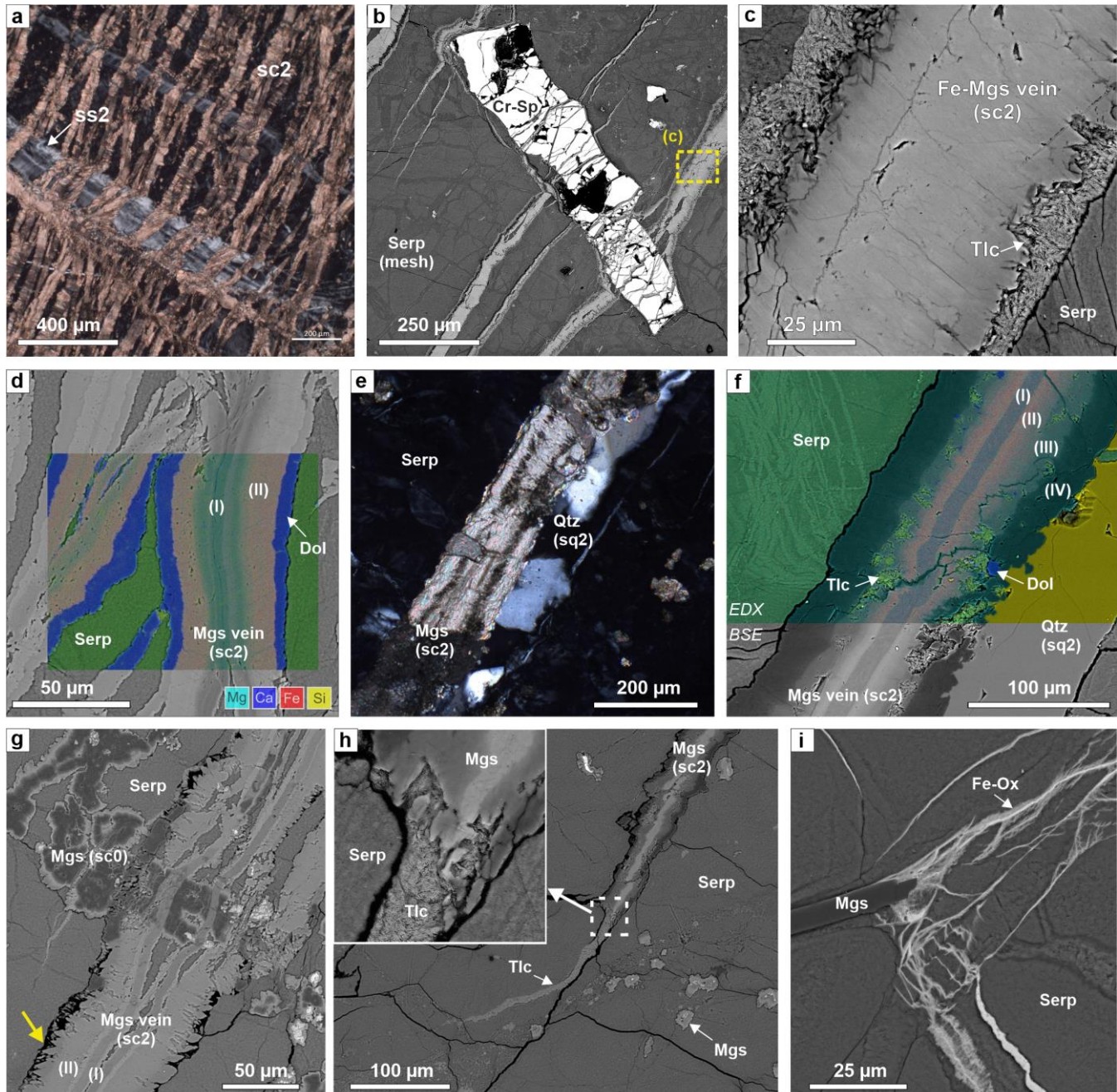

**Figure 4: Microstructures of zoned magnesite veins (sc2) in serpentinite.** (a) close spacing of carbonate; the veins are deflected where they cut a crack-seal serpentine vein (xpol). (b) Elongated Cr-spinel and serpentine mesh cut by zoned, Fe-magnesite-talc veins (BSE image). (c) Detail of b, showing talc seams along vein walls. (d) Composite-color EDX map superposed on a BSE image of zoned sc2 veins with dolomite rims (I: Ca-bearing magnesite; II: Fe-magnesite); phase legend see Fig. 3c. (e) Quartz vein exploiting the vein-host interface of a zoned magnesite vein (xpol). (f) Composite-color EDS map superposed on a BSE image of a part of the vein in e), with zones: (I) Ca-bearing magnesite, (II) Fe-magnesite, (III) Fe-bearing magnesite with talc inclusions, (IV) Fe-poor magnesite. Small Mg/Si variations show thin serpentine veins in the matrix. (g) Complex cross-cutting relations between zoned sc2 vein and earlier mesh-pseudomorphic magnesite (sc0), showing that the core of sc2 veins formed by opening while the vein rims are micro-reaction fronts, with euhedral crystal terminations at the vein walls (yellow arrow). (h) Tip of zoned sc2 carbonate vein in serpentinite. At the termination of magnesite, a talc veinlet continues over a short distance. This talc veinlet does not correspond to the open micro-cracks in serpentinite (black lines), showing that those are later and not related to vein formation of the zoned magnesite vein. (h) Fe-oxides/hydroxides veinlets emanating from magnesite vein tip. (a, d, g: BT1B_44-2_47-50; b, c, e, g, h: BT1B_44-3_9-11; i: BT1B_39-3_9-13)

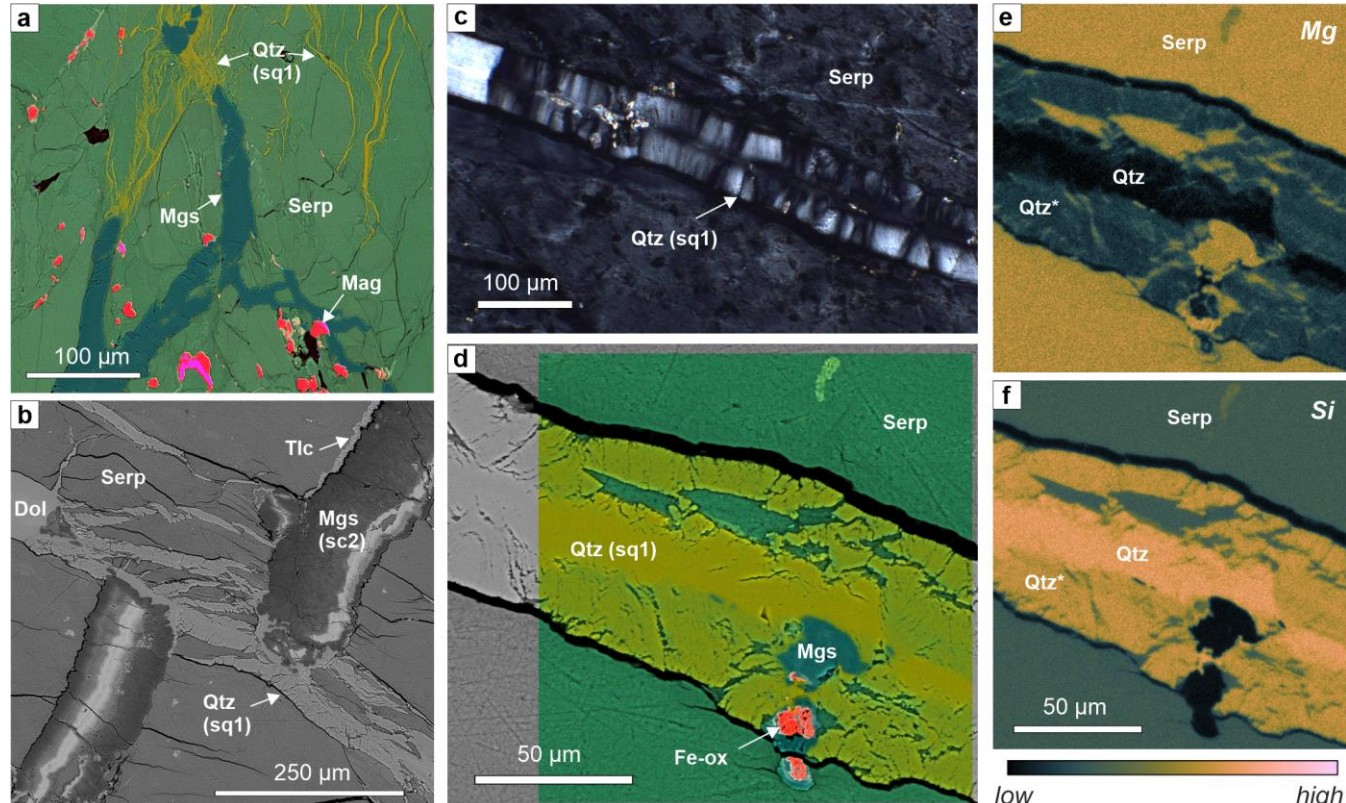

**Figure 5: Quartz veins in serpentinite of Hole BT1B. (a)** Composite-color EDX map of feathery quartz micro-veins with interstitial serpentine emanating from magnesite vein tips (sample BT1B_39-3_9-13). **(b)** BSE image of branched quartz-vein with minor dolomite cutting and offsetting zoned magnesite vein. **(c)** Fibrous, antitaxial microstructure of quartz vein in serpentinite (xpol). **(d - f)** Composite-color EDX map superposed on BSE image, and corresponding Mg- and Si maps of a quartz vein in serpentinite (common color scale shown below panel f), showing that only the median zone consists of pure SiO₂, while the vein walls (Qtz*) contain a high amount of serpentine and, locally, carbonate (nano-) inclusions, which result in the overall chemistry of Mg-bearing quartz at the spatial resolution of the EDX measurements. **(b-f: sample BT1B_44-3_9-11)**

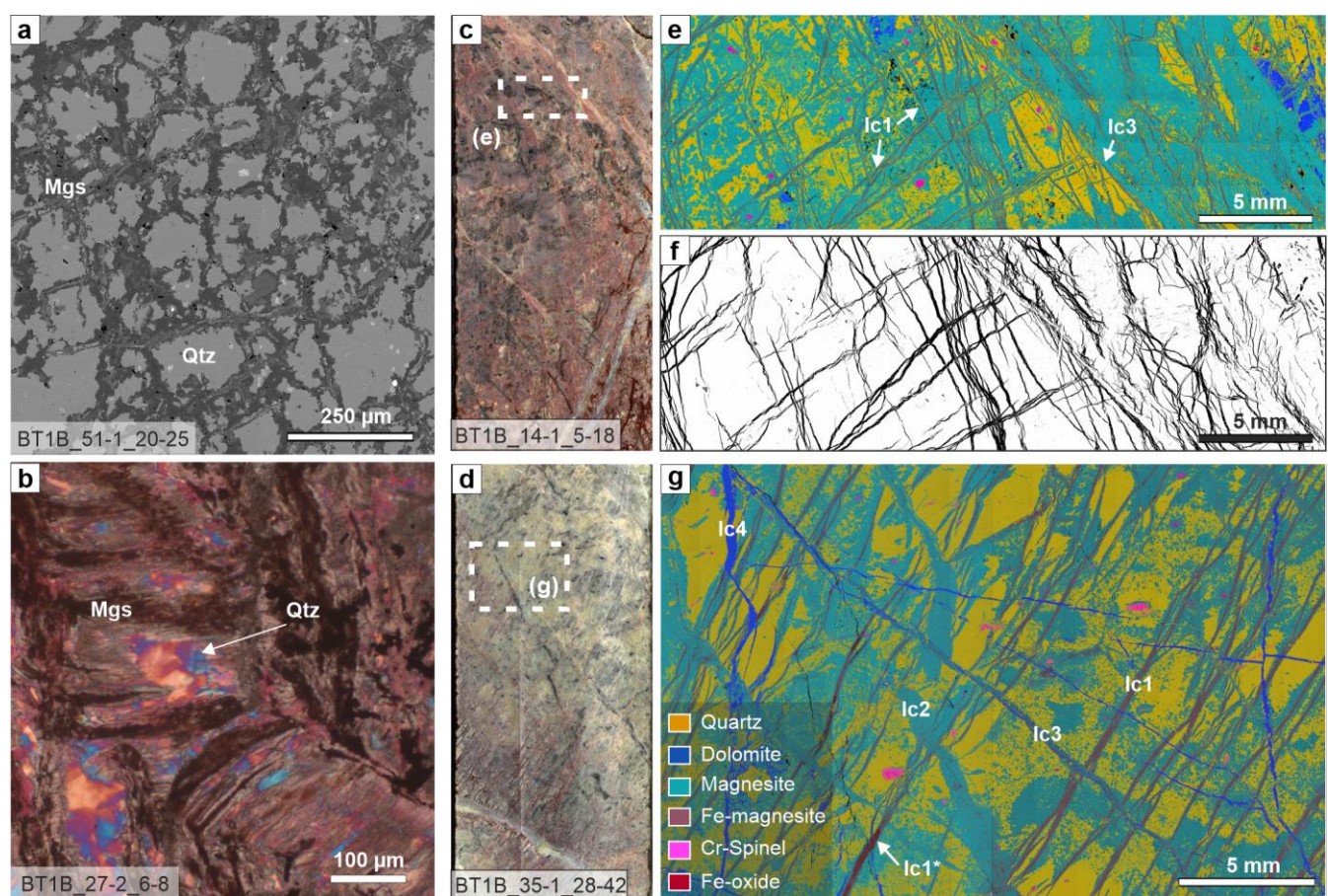

**Figure 6: Veins in listvenite of Hole BT1B. (a) Magnesite pseudomorphic mesh vein network ($l_{ss0}$, BSE image). (b) Pseudomorphic magnesite-quartz after serpentine crack-seal vein ($l_{ss2}$, xpol with 1λ-plate) (c) Split-core image of listvenite with conjugate cross-cutting and locally anastomosing zoned magnesite vein network. (d) Split-core image of listvenite with parallel carbonate veins that are locally oxidized. (e) Composite-color EDX map of the indicated area in c), showing different magnesite vein generations, and a patchy quartz distribution in the matrix (common legend in g). (f) Same area as in e), segmented for Fe-magnesite (15.8 %). (g) Composite-color EDX maps of the area in d) showing cross-cutting relationships between different carbonate vein generations (c.f. Table 1). Dark-red veins ($lc1^*$) contain abundant Fe-oxides (red veins in d). Core diameter in a) and d) is 6.3 cm.**

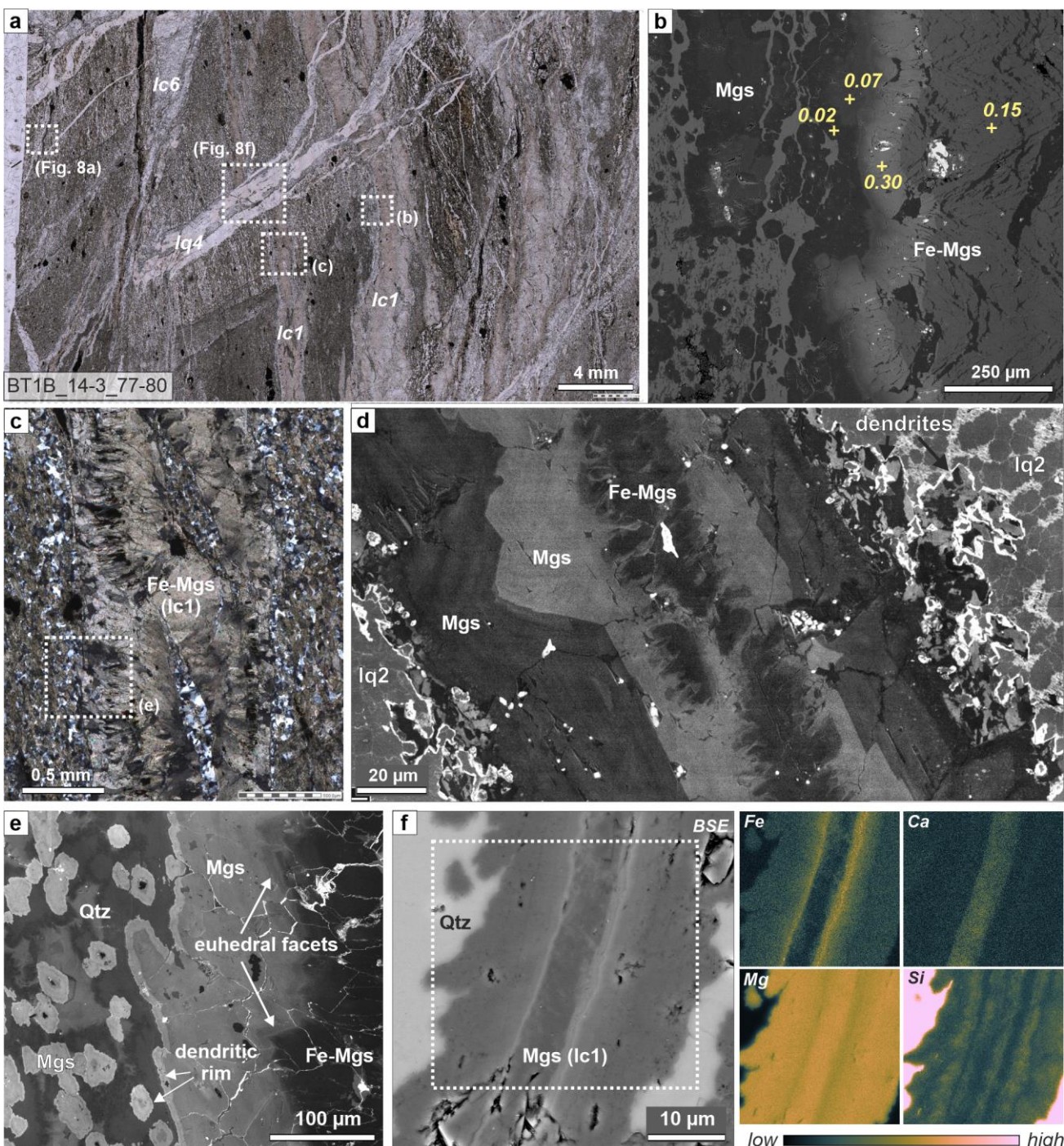

**Figure 7: Zoned magnesite veins in listvenite. (a) Thin section overview of listvenite with comparatively wide, zoned Fe-magnesite veins (lc1), cut by a quartz-magnesite vein (lq3). (b) BSE image of a rim of a zoned magnesite vein, with Fe/(Fe+Mg) of magnesite shown for different zones (from EDX spot measurements). (c) Crossed-polarized micrograph of zoned magnesite vein with lens-shaped host listvenite inclusions. (d) Pan-chromatic SEM-CL image of lc1 magnesite vein with Fe-rich median zone, concentrically zoned growth with euhedral facets towards the vein walls, and dendritic boundaries with bright-luminescent $SiO_2$ overgrowth. The matrix consists of micro-crystalline quartz (lq2) (sample BT1B_67-2_36-40). (e) SEM-CL image of the area indicated in c), showing euhedral magnesite growth towards the vein walls. (a, b, c, e: sample BT1B_14-3_77-80). (f) BSE image and EDX maps of the dotted area of thin zoned magnesite vein in listvenite, showing systematic variations in the amount of $SiO_2$ nano-inclusions that are apparent as variable Si-bearing magnesite at the resolution of EDX measurements (sample BT1B_27-2_6-8). Contrasts are adapted for each EDX map separately to improve visibility of zoning.**

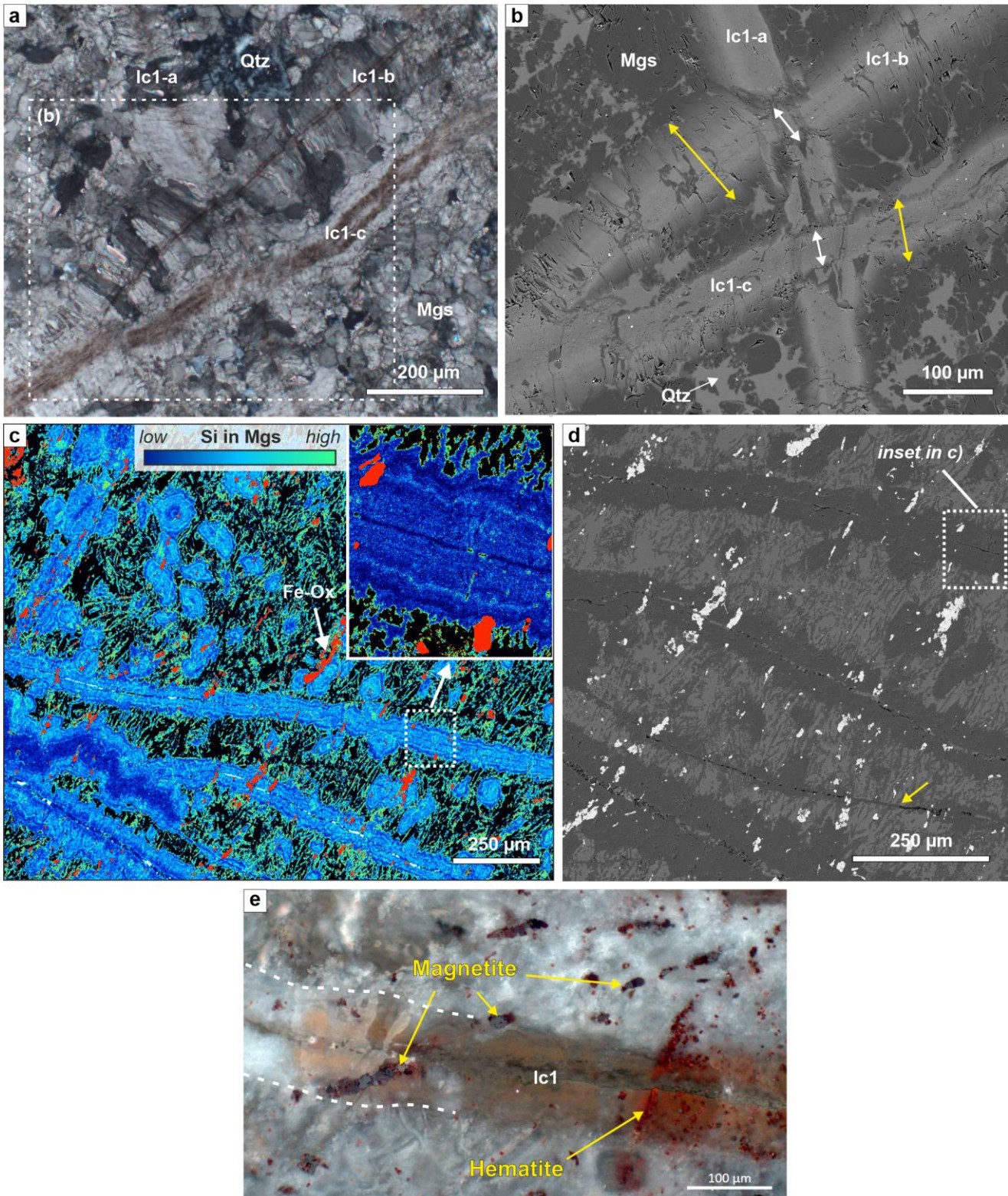

**Figure 8: Microstructures showing the extent of opening versus host rock replacement by zoned magnesite veins. (a)** Different generations of zoned magnesite veins with brown, Fe-enriched median zone, showing fibrous (lc1-a, lc1-b) and wide-blocky (lc1-c) habits (xpol). **(b)** BSE image of the area marked in a), showing that the true vein aperture via opening at the intersection with the earlier vein lc1-a corresponds to inclusion-bearing median zones only (white arrows), while fibrous to euhedral overgrowths indicate that a significant fraction of the total vein thickness (yellow arrows) must have formed due to replacement of the host rock (BT1B_14-1_7-11). **(c)** Si content in magnesite from EDX mapping (with Fe-oxides in red; quartz - black; minor dolomite - white), showing vein-parallel zoning in carbonate veins and concentric zoning in matrix magnesite ellipsoids. Only the median zone of veins cut the oblique Fe-oxides. Local folding of an Si-poor vein core in the lower left. **(b)** BSE image of part of the area in c); the yellow arrow marks a thin vein that did not develop the common replacement rim. **(e)** Crossed-polarized, reflected light image of relationship between wide-blocky zoned magnesite vein (lc-1), magnetite and hematite. (c, d & e: BT1B_21-3_35-40).

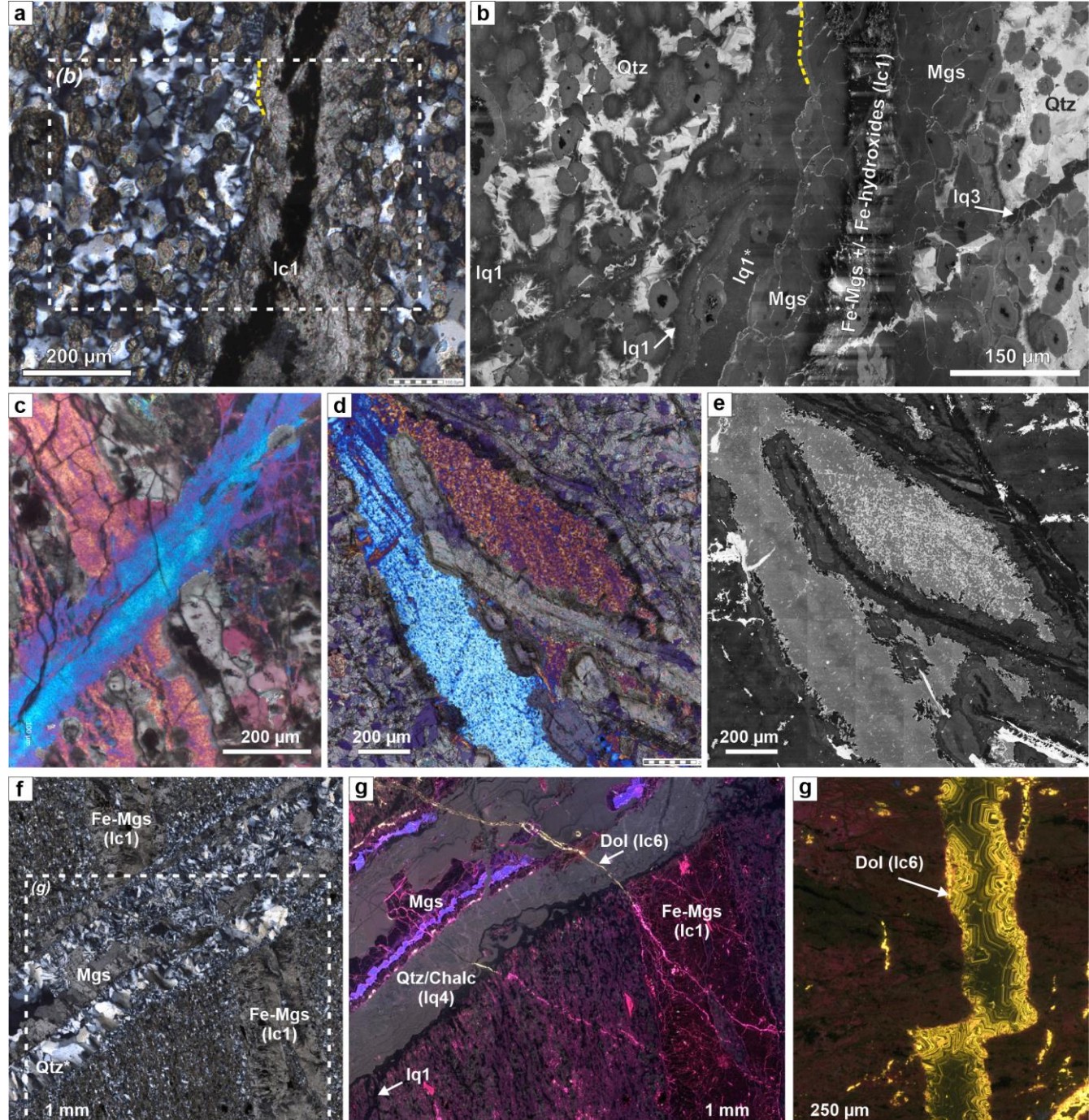

**Figure 9: Quartz and late dolomite veins in listvenite. (a)** Crossed polarized micrograph of cryptic quartz vein adjacent to a zoned magnesite vein (lc1). **(b)** SEM-CL image of the marked area in a), showing two generations of dark luminescent cryptic quartz veins (lq1 and lq1*) in bright luminescent matrix quartz and exploiting the zoned magnesite vein interface. Cryptic quartz veins do not cut magnesite ellipsoids in the matrix. A later quartz vein (lq3) cuts both magnesite ellipsoids and the zoned magnesite vein. **(c)** Two crosscutting generations of micro-crystalline quartz veins with distinct crystal preferred orientations (xpol with 1λ-plate). **(d & e)** Crosscutting relationship between zoned magnesite veins and micro-crystalline quartz (d: ViP xpol with 1λ-plate; e: SEM-CL); dendritic magnesite overgrowths on the carbonate vein are undisturbed by the micro-crystalline quartz. **(f & g)** Crossed polarized micrograph and optical CL mosaic image of a quartz/chalcedony-magnesite vein (lq4) cutting a zoned magnesite vein. Magnesite in the vein has distinct luminescence from that in the listvenite matrix. A late, thin dolomite vein cuts all other vein generations. **(g)** Optical CL mosaic image of a late syntaxial lc6 dolomite vein in listvenite, showing dull to bright yellow oscillatory growth zoning into an open fracture. The vein is roughly oriented vertical (parallel with respect to the core orientation). (a, b, f, g: sample BT1B_14-3_77-80, see marked areas in Fig. 7 a; c: BT1B_56-4_45-50; e, f: BT1B_67-2_36-40; g: BT1B_32-1_17-19).

**(II, III) Serpentinization | + $H_2O$**

**(IV) Early carbonate | +$CO_2$ | −$SiO_2$***

**(VI, VII) zoned growth | +$CO_2$ | −$SiO_2$***

**(VIII) dendrites & quartz | +$CO_2$, +$SiO_2$***

**(IX) quartz-carbonate veins | +$SiO_2$**

**(X) cataclasis, late veins | +Ca | −Mg***

Legend:
- olivine/brucite
- serpentine veins
- early magnesite (sc0; sc1)
- Fe, Si zoned magnesite
- quartz
- serpentine (mesh/foliated)
- banded serpentine crack-seal vein
- early carbonate (median zones of sc2)
- dendritic magnesite
- late magnesite/late dolomite

**Figure 10:** Sketch of evolution of carbonation and vein formation in Hole BT1B, as inferred from serpentinite and listvenite samples (different stages see text). Different stages were related to differing element transfer with gains (denoted by e.g. + $H_2O$); inferred losses (e.g. −$SiO_2$*) are likely only relevant on a local scale. For clarity, not all vein generations (c.f. Tables 1 & 2) are shown.

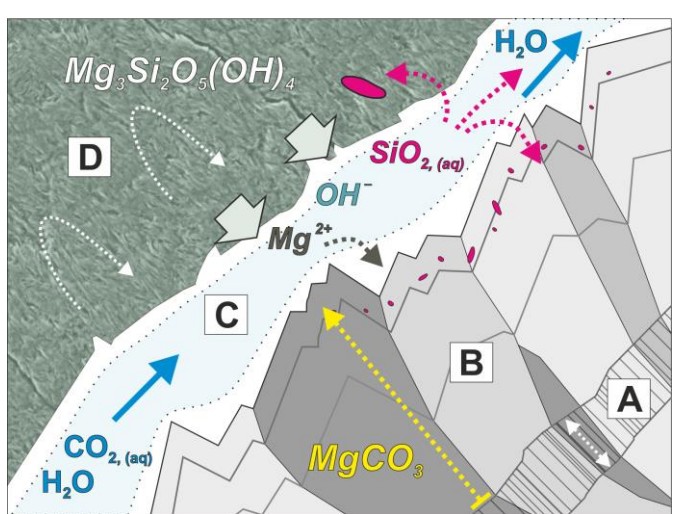

**Figure 11:** Conceptual sketch of magnesite vein growth by replacement of serpentine. Domain A: initial dilatant fracture with carbonate infill, forming the median lines of zoned magnesite veins. B: magnesite vein growth rim formed by replacement. C: interface permeability at vein walls causes development of a fluid film with diffusion-dominated boundary layers (white) and a central advective flow zone (light blue), which allows $CO_2$ influx and drives coupled serpentine dissolution and carbonate precipitation. The widths of boundary layers relative to the advective zone depend on fluid flow rate. The overall width of domain C is highly exaggerated for illustration purposes. D: nano-porous serpentine with predominantly diffusive solute transfer in matrix permeability. The observations show that only small fractions of dissolved silica precipitate in situ (red), forming nano-inclusions in magnesite; most silica is leached and precipitates in other micro-environments.

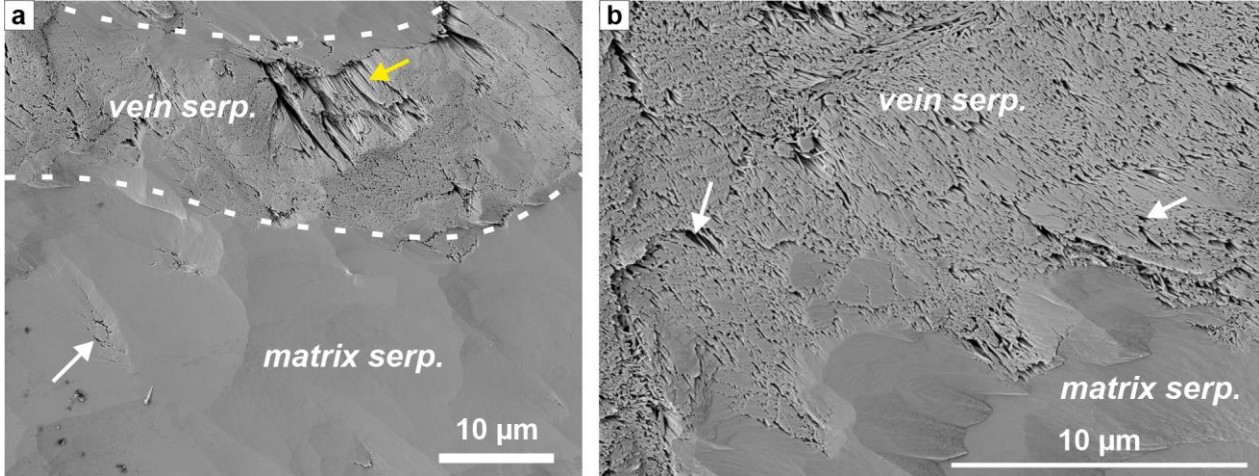

**Figure 12: SE image of a broad-ion-beam polished serpentine vein in foliated serpentinite (a) with a close-up of the vein-wall rock interface (b), showing that the vein serpentine is highly (nano-)porous in comparison to matrix serpentine. White arrows indicate examples of nano-porosity (black). In the vein, porosity is controlled by fibrous serpentine as seen in the poorly polished area (yellow arrow in a). The vein domain corresponds to the blue bands in Fig. 2d (sample OM20-13).**

**Tables**

**Table 1: Vein classification in Hole BT1B serpentinites (ordered from relatively older to younger)**

| ## | Vein type | Sub-types | Characteristics | Mineralogy | Width / *adundance* | Examples in thin section |
|---|---|---|---|---|---|---|
| ss0 | serpentine mesh ("serp mesh") | | brownish serpentine (± Brc?) forming a polygonal network, often with magnetite in median zone of veins | Serp, Mag, ± Brc? | 3 – 20 µm *ubiquitous* | most non-foliated serpentinites |
| ss1 | serpentine-magnetite* | various generations | transparent serpentine with magnetite aggregates in median zone | Serp (Lz), Mag | 0.1 – 1.5 mm; *minor* | BT1B_74-1_77-80 |
| ss2 | serpentine crack-seal ("serp") | various generations | cross-fiber crack-seal veins with banded extinction patterns; pull-apart structures are common; locally parallel sets, en-echelon | alternating Chr-Lz intergrowths? (Tarling et al., 2021) | 50 – 500 µm *common* | BT1B_39-3_9-13 BT1B_44-2_47-50 BT1B_74-1_77-80 |
| sc0 | pseudomorphic carbonate* | mesh carbonate carbonate after ss2 | carbonate aggregates along the serpentine mesh or parallel to fibers of serpentine crack-seal veins; locally euhedral facets at contact to serpentine | Mgs (variably Fe-bearing); ± Dol | 5 – 30 µm *common* | BT1B_44-2_47-50 BT1B_44-3_9-11 |
| sc1 | cleavage-parallel carbonate* | | patchy to feathery, in places dendritic vein aggregates parallel to serpentine cleavage, locally folded | Fe-poor Mgs ± Dol; Dol ± Qtz in some samples | 0.1 – 1 mm *minor* | BT1B_44-2_47-50 BT1B_44-3_9-11 BT1B_42-1_19-24 |
| sc2 | zoned magnesite („carb-oxy") | - anastomosing magnesite - composite Fe-magnesite – talc - parallel Fe-magnesite-dolomite | antitaxial / fibrous, with Fe-oxide bearing median line and bisymmetric chemical zoning; host serpentine inclusions are common | common zoning: median line: locally Fe-(hydr)oxide; zoned vein core: Ca-bearing, Fe-rich Mgs, with variable Si-inclusion content; rims: Fe-poor Mgs, Dol | typically 50 – 200 µm; *very common in some intervals* | BT1B_39-2_34-36 BT1B_39-3_9-13 BT1B_39-4_14-18 BT1B_44-2_47-50 BT1B_44-3_9-11 |
| sc3 | magnesite | | irregular; cross-fiber to blocky | mostly Fe-poor Mgs, locally Mn-bearing | 200 – 500 µm; *minor* | BT1B_39-4_14-18 |
| sq1 | feathery quartz* | locally 2 or more generations | thin splayed / feathery, highly irregular quartz vein aggregates, partly antitaxial, locally emerging from magnesite vein tips | Qtz ± Mgs, Dol in parts: impure Qtz with nm-µm Serp inclusions and intergrowths | < 50 µm *minor* | BT1B_39-2_67-72 BT1B_39-3_9-13 BT1B_42-2_19-24 BT1B_44-3_9-11 |
| sq2 | quartz / quartz-magnesite | | granular – blocky quartz, locally vuggy and with euhedral magnesite | Qtz, ± Fe-poor Mgs | 0.1 – 10 mm *minor* | BT1B_44-3_9-11 BT1B_44-2_47-50 |
| sc4 | late carbonate | | often brecciated, partly vuggy | Dol and/or Mgs | 0.1 mm – > 5 cm | |

*Mineral abbreviations: Mgs – magnesite, Dol – dolomite, Qtz – quartz, Tlc – talc, Serp – serpentine, Lz – lizardite, Chr – chrysotile, Brc – brucite, Mag – magnetite, Hem – hematite, CrSp – Cr-spinel, FeOx – Fe-oxide/hydroxide. * = cross cutting relationships (relative timing) ambiguous; thin section samples are named with an abbreviated form of the ICDP convention, following the scheme "Hole_Core-Section_top-bottom [cm]".*

**Table 2: Vein classification in Hole BT1B listvenites (ordered from relatively older to younger)**

| ## | Vein type | Sub-types | Characteristics | Mineralogy | Width / *abundance* | Examples in thin section |
|---|---|---|---|---|---|---|
| $l_{ss0}$ | magnesite or quartz network after serpentine mesh | - magnesite network<br><br>- quartz network | in mesh-pseudomorphic listvenites; vein network discontinuously follows prior polygonal serpentine mesh | variably Fe-bearing Mgs (± Hem, Mag relicts); Qtz | 3 – 20 µm<br>*locally ubiquitous* | BT1B_21-3_35-40<br>BT1B_51-1_20-25<br>BT1B_55-3_68-72<br>BT1B_56-1_55-60 |
| $l_{ss2}$ | magnesite-quartz after serpentine crack-seal | | pseudomorphic replacement of serpentine veins by vein-perpendicular magnesite columns + quartz | Mgs, Qtz | 50 – 500 µm; *minor* | BT1B_27-2_6-8 |
| lc1 | zoned magnesite („carb-ox veins") | - anastomosing<br>- parallel<br>- cross-cutting (two generations, or conjugate) | fibrous antitaxial to wide blocky; may define a macroscopic foliation where abundant; irregular vein walls. Locally folded and transposed by matrix foliation. | median line: locally Fe-(hydr)oxide, Dol or Qtz; zoned vein core: Ca-bearing, Fe-rich Mgs, with variable Si-inclusion content; rims: Fe-poor Mgs, Dol | 50 µm to locally up to 1 mm<br>*very common* | BT1B_14-1_7-11<br>BT1B_14-3_77-80<br>BT1B_16-3_28-31<br>BT1B_20-1_64-68<br>BT1B_21-3_35-40<br>BT1B_31-4_12-14 |
| lc2 | magnesite ("carb") | | irregular; cross-fiber to blocky syntaxial | Fe-poor Mgs (dull/non-luminescent) | 20 – 500 µm; *minor* | BT1B_14-1_7-11<br>BT1B_14-3_77-80 |
| lc3 | magnesite-dolomite | | irregular; polycrystalline | Mgs (bright pink luminescent), ± Dol | 10 – 200 µm; *minor* | BT1B_14-3_77-80 |
| lq1 | cryptic quartz | | matrix veins with irregular walls, not cutting magnesite ellipsoids; often only visible in CL | Qtz (dull luminescent) | 10 – 100 µm | BT1B_14-3_77-80 |
| lq2 | microcrystalline quartz* | up to two generations | irregular to patchy, polygranular with domains with strong CPO; not cutting magnesite ellipsoids | Qtz (after opal?) | 50 µm – 1 mm; *minor* | BT1B_56-4_45-50<br>BT1B_67-2_36-40 |
| lq3 | quartz | | straight veins cutting magnesite ellipsoids / cutting pseudomorphic mesh | Qtz (dull and bright luminescent), ± carbonate | 10 – 100 µm; *minor* | BT1B_14-3_77-80 |
| lq4 | magnesite-quartz ("carb-qtz") | | syntaxial; branched; host rock inclusions common; straight vein walls; polycrystalline and radial chalcedony / quartz aggregates; often magnesite in the vein center | Qtz / chalcedony (bright luminescent), Fe-poor Mgs | typically > 1 mm<br>*common* | BT1B_14-3_77-80<br>BT1B_20-1_64-68<br>BT1B_51-1_20-25 |
| lc4 | quartz-dolomite | | syntaxial, commonly with euhedral facets; host rock inclusions common; straight vein walls; often carbonate in the vein center | Qtz, Dol | 0.1 – 1 mm<br>*common* | BT1B_21-3_35-40 |
| lc5 | late magnesite | | syntaxial / polygranular; irregular, with host listvenite inclusions; yellowish in drill core | Fe-bearing Mgs (not zoned) | > 2 mm<br>*minor* | BT1B_68-3_60-65 |
| lc6 | late dolomite (various generations) | - thin en echelon<br>- brecciated<br>- vuggy | syntaxial to blocky, planar with straight vein walls; in parts brecciated; partly open (vuggy) | mostly Dol, with oscillatory growth zoning in CL | 0.1 mm – > 5 cm<br>*common* | BT1B_14-3_77-80<br>BT1B_16-3_28-31<br>BT1B_20-1_64-68 |

*Mineral abbreviations: Mgs – magnesite, Dol – dolomite, Qtz – quartz, Tlc – talc, Serp – serpentine, Lz – lizardite, Chr – chrysotile, Brc – brucite, Mag –*
*magnetite, Hem – hematite, CrSp – Cr-spinel. Numbering of vein generations: subscripts denote that veins are pseudomorphic replacements of previous*
*serpentine vein generations. * = cross cutting relationships (relative timing) ambiguous; thin section samples are named with an abbreviated form of the*
*ICDP convention, following the scheme "Hole_Core-Section_top-bottom [cm]".*
