# Peer review of "Progressive veining during peridotite carbonation: insights from listvenites in Hole BT1B, Samail ophiolite (Oman)"

_Solid Earth, 2021_

## Author Comment (AC1)

We thank Dr. Dennis Quandt and the anonymous reviewer for their constructive and detailed reviews, which helped us to improve several aspects of the paper. In the attached author comments letter, we addressed each comment separately and explain how we implemented improvements. The reviewer's comments are italicized, followed by our point-by-point response in blue.

**RC1** (Dennis Quandt)**

**General comments**

In this manuscript the authors present detailed petrographic data and element mappings of veins in order to infer the processes of serpentinite carbonation. They establish a model on veining that may be of interest to the vein and serpentinization community. In my opinion this manuscript requires major revisions before it can be considered for publication. My main suggestions for improvements concern (a) the clarification of the descriptive part and (b) some reorganization of the discussion. I also think that (c) more emphasis could be put on the tectonic framework in which the veins formed.

(a) With ca. ten vein types in each host rock lithology (see table 1 and 2) and without a schematic figure illustrating the vein mineralogy, microtextures, and their spatial relationships, this manuscript is difficult to understand. Therefore, I recommend to include a figure that clearly shows the different vein types. This should be part of the results chapter.

We have carefully thought about this recommendation and decided against including such a schematic figure, because we think that it would be highly redundant to the information contained in Tables 1 and 2, while being less precise.

(b) In several parts of the discussion, an idea/model/interpretation is presented followed by a description of supporting petrographic observations. In order to enhance the comprehensibility, the observations should be stated in a short phrase and then discussed, not vice versa. Very long phrases should be shortened. Apart from that the manuscript is well written.

We agree that in section 5.4 the order may have been confusing, we modified it to state observations first. In the other parts of the discussion we do not think that changing the order would enhance comprehensibility; we believe that a statement of a key observation and related hypothesis in the beginning, followed by a discussion about which data support or contradict this hypothesis is a clear and comprehensible structure. In particular in some cases (e.g. explanations of syn- vs. antitaxial veining, reaction-induced fracturing) we deem it necessary to outline the general models at the beginning of the discussion sections, before discussing the observed microstructures in their light, because not all readers will be familiar with these concepts.

We have shortened long phrases in various places during revision of the manuscript.

(c) In the discussion, veins are interpreted to be associated with tectonic stresses. For this purpose, the regional geological framework could be taken into account in greater detail. Moreover, if there is enough data on listvenites from other settings, listvenite formation/veining in different ophiolites/tectonic settings could be briefly compared. This might be also the basis to test the models presented here.

We have added a paragraph discussing the tectonic framework and how it links to veining. While veins are extremely common in listvenites worldwide, this is to our knowledge the first detailed microstructural investigation of vein formation mechanisms in natural listvenites. Detailed studies of veining exist for classical syntaxial, antitaxial and blocky veins in various lithologies, but as they usually are not related to reaction-driven cracking +/- replacement it is difficult to compare these processes directly.

**Specific comments**

L. 32: The topic of carbon sequestration is mentioned here; can this idea be picked up again in the discussion/conclusion? Are there implications of your study for carbon sequestration?

Thank you for this suggestion. We add to the conclusions: "Our results suggest that in this natural example, veining caused by tectonic stress and fluid overpressure is an important mechanism to create permeability despite carbonate precipitation. Without the added effect of tectonic deformation and related deviatoric stress, it is possible that permeability created through reaction-driven fracturing +/- replacement veining alone is not enough to allow for the necessary fluid-flux for carbonation to progress. Therefore, the extent of carbon mineralization and permanent CO2 sequestration that can be attained via experimental in-situ CO2 injection might be limited."

*L.* 42: "Reproducing conditions of listvenite formation at a large scale is experimentally challenging [...]" partly repeats L. 37: "[...] experiments have so far not been able to reproduce this reaction [...]". Merge them to one phrase.

We modify L. 42 so that it is less repetitive.

L. 66: "relative timing" instead of "timing"

implemented

L. 74-76: I have the impression that the more recent literature favors a supra-subduction zone over a mid-ocean ridge setting. Is that true? With regard to the general comment (c), a more detailed description might be required.

Yes, the more recent literature is more in favour of a supra-subduction zone setting. We modify this line to include this information, although this question is not directly relevant to the findings of our study.

L. 74-77: As I understand, the term "ophiolite crystallization" here refers to the formation of the mantle rock sequence. However, a complete ophiolitic sequence representing obducted/uplifted oceanic crust may also contain sedimentary rocks on top that did not crystallize. Therefore, consider to change the term "ophiolite crystallization".

Changed to "crystallization of the oceanic crust"

L. 91-150: This section partly gives the impression that it is a results chapter. Indeed, the last phrase of this section "In this study, we refine the preliminary vein classification [...]" clears this up, but a general phrase in the beginning shortly stating what has been published on this topic would give the section a better structure in my opinion. Also consider to move some general aspects to chapter 2.1.

Yes, good point. We add a statement at the beginning of section 2 to clarify that the sections 2.1, 2.2, and 2.3 contain an overview of the geological setting and a summary of results from previous studies.

L. 100-110: Consider to restructure this section as follows: first the old models followed by the new models.

Implemented.

*L.* 185-onward: In addition to a figure summarizing the vein types, also consider to consistently mention the vein abbreviations as given in Table 1 and 2.

We added consistent mentions of vein abbreviations in each sub-section.

L. 210-376: Results chapter: There are around ten different vein types described in each lithology. It would considerably help to provide a figure that shows a schematic overview of the different vein types. Among others, this should include vein type, mineralogy, host rock, and crosscutting relationships. Also consider if the different vein types can be merged in order to simplify the structure. Please see our reply to major comment (a).

We have already merged different veins into main types (see sub-types in Table 1 & 2). We think it is not a good idea to further simplify this because that would result in a loss of information and may foster an over-simplistic view of the involved processes; in contrast, we observe that veining in the BT1 listvenites is highly variable suggesting a complex interplay of changing processes and conditions throughout progressive fluid-rock interaction.

L. 249: "surprisingly" sounds subjective.

We change the wording to avoid subjective terms.

L. 377-onwards: are the characteristics of drill core samples and fieldwork samples comparable? Any differences that may indicate localized processes etc.?

The drill core samples and field observations and samples are comparable. We focus our analysis here on the samples of BT1 because the sample set is larger and thus more representative, and because structural relationships are only rarely well visible in the field due to a thick orange-red weathering rim. We add the following statement to the end of section 5.1 to clarify the heterogeneity and that vein structures in calcite-dolomite listvenites, which are not studied in detail here, may be somewhat different: "We note that not every stage I - X is present in each core section. Moreover, field exposures near site BT1 show high variability with strongly veined domains alternating with vein-poor, massive listvenite intervals. We further note that vein microstructures in dolomite-calcite listvenites that are common further north in the Fanjah region are somewhat different from the magnesite-dominated BT1B listvenites studied here."

L. 400: "Incipient carbonate precipitation as ellipsoidal/spheroidal grains in the serpentine matrix [...]"; is this carbonate the same as sc0 in Figure 10? If yes, mention sc0 in the main text.

We add an indication to sc0 when referring to Fig. 2, Fig. 10 in this sentence.

L. 418: Syntaxial veins: an important characteristic of syntaxial veining is growth competition. I could not find that the term "growth competition" was mentioned in the text. Is this because there is no growth competition?

The syntaxial veins indeed show abundant signs of growth competition, we add this observation to section 4.2.6.

L. 427: Change "[...] steps (4) - (8) may have occurred [...]" to "[...] steps IV-VIII may have occurred [...]" in order to be consistent.

done

L. 471-472: "Current models of vein formation treat the host rock as a non-reactive substrate with vein formation due to precipitation from aqueous solution in fluid-filled fractures [...]"; this probably represents an important point by which this manuscript stands out from other recent publications on the same/similar topic. If this is the case, also consider to mention the process of "replacement veining" in the last paragraph of the introduction.

Yes, that is true, this is a fundamental difference from classical vein formation models – we modify the last sentence to emphasize replacement veining.

L. 482: My understanding is that during antitaxial veining, outward growing mineral fibers are in contact with the host rock (i.e., force or pressure of crystallization). Therefore, I would not expect "significant permeability along the vein-host rock interface" as stated in the text. It is also difficult to compare permeability of different vein types without defining fracture or vein aperture, mineral growth rate etc.

As outlined in the paragraph before L. 482, we do not consider these zoned veins as classical antitaxial veins, but as replacement veins (in addition to dilatancy) with an apparent antitaxial habit.

Significant permeability along the vein-host rock interface is required for the replacement to proceed as inferred; with "significant" we mean here: enough permeability to allow CO2-H2O fluid influx to trigger serpentine dissolution, carbonate precipitation, and partly SiO2,aq leaching along this interface. We acknowledge that the usage of the term antitaxial in this sentence may have been misleading, we replaced it with "carbonate vein growth by replacement from the center of the vein toward the wall rock.".

L. 483-484: "[...] fracture permeability created initially by dilatant opening of the vein, which may easily clog due to mineral precipitation [...]"; is that also true for slow vein mineral growth rates? See also comment above.

We have no constrains on vein growth rates. Experiments and isotope studies have shown that carbonation of ultramafic rocks can proceed very fast (e.g. Beinlich et al., 2020a, Nature Geoscience, doi: 10.1038/s41561-020-0554-9). Dendritic growth structures onto vein margins (see Fig. 7d) as well as on the rims of matrix magnesite grains suggest that at least some growth stages were fast, while euhedral growth steps may have formed at lower rates. It is true that antitaxial veins with slow growth rates may not clog, but such a simplistic model is not consistent with our observations, which indicate a dynamically evolving reactive system with different precipitation rates, serpentine dissolution kinetics (which would vary depending on CO2 flux) and carbonate growth kinetics.

L. 485-490: Are there chemical gradients from vein to host rock that corroborate your interpretation of a reactive interface between vein and host rock, i.e., element depletion in the host rock close to the vein and corresponding element enrichment in the vein minerals indicating reactions?

We did not observe element depletion rims in the host rock close to the veins, such features might become apparent by doing trace element analysis / mapping. But there is other evidence clearly indicating reaction, which we laid out in the discussion of section 5.3 (see also Fig. 8) and which we consider to be strong arguments for serpentine replacement along the vein – host rock interface: (i) the magnesite veins in question (first of all sc1, sc2 and lc1 veins) commonly contain secondary silica phases that are typical reaction products of the reaction of serpentine with CO2 (quartz, and in places talc; although less than expected, as explained now in more detail in section 5.1), (ii) the growth zoning patterns are equivalent to the zoning of magnesite growing in the matrix, and (iii) the replacement parts of these veins passively overgrow markers such as Cr-spinel, magnetite and previous magnesite veins. The full replacement of serpentine – and not only leaching of some elements from the host rock – during reaction with CO2 is typical for listvenite formation, and also commonly observed in experiments (see e.g. Sieber et al., 2018).

Sieber, M. J., Hermann, J., and Yaxley, G. M.: An experimental investigation of C–O–H fluid-driven carbonation of serpentinites under forearc conditions, Earth and Planetary Science Letters, 496, 178-188, 2018)

L. 497-500: Growth zonation in calcites may be also caused by varying growth rates in association with alternating Mn incorporation. Is this model applicable to your observations? Moreover, check if geochemical self-organization (autonomously developed patterns in a closed system without external control) may apply here as a cause for zoning patterns, especially if the patterns are highly oscillatory. The following references may be of interest:

Dromgoole, E. L., & Walter, L. M. (1990). Iron and manganese incorporation into calcite: Effects of growth kinetics, temperature and solution chemistry. Chemical Geology, 81(4), 311-336.

Reeder, R. J., Fagioli, R. O., & Meyers, W. J. (1990). Oscillatory zoning of Mn in solution-grown calcite crystals. Earth-Science Reviews, 29(1-4), 39-46.

**Wang, Y., & Merino, E. (1992). Dynamic model of oscillatory zoning of trace elements in calcite: Double layer, inhibition, and self-organization. Geochimica et Cosmochimica Acta, 56(2), 587-596.**

Thank you for this interesting comment. Magnesite growth rates may indeed be influenced by variable Fe and Mn incorporation, similar to calcite. However, we think that the models for calcite are not fully applicable for the magnesite veins: rather than highly oscillatory zoning patterns, the Fe content and CL images show variable but systematic trends in magnesite composition with progressive growth (e.g. Fig. 4f, Fig. 7), which in places is complementary to variable incorporation of inclusions of silicate reaction products (talc in a few places, and commonly quartz). Similar growth zoning in magnesite related to different Fe-, Mn- and Ni- partitioning into reaction products during progressive carbonation has been documented in listvenites elsewhere (e.g., Menzel et al., 2018 Lithos; Tominaga et al., 2017 Nature Communications) (albeit for matrix magnesite grains and not veins). Therefore, we consider the zoning patterns to be primarily the result of different reaction affinity of CO2 with magnetite, Fe-serpentine, Fe-poor serpentine and (possibly in places) talc within the reacting rock volume, which releases different amounts of Fe, Mn and Ni to be incorporated into magnesite throughout the progressive carbonation reaction. Fluctuations in CO2 concentration, pH, and oxygen fugacity of the infiltrating fluid as well as variable flux rates – in parts as a consequence of reaction progress – will strongly influence the magnesite growth rate and thus also the rate of incorporation of minor and trace elements, but we consider it very unlikely that geochemical self-organization alone would produce the patterns observed here.

We observe oscillatory zoning patterns in late dolomite veins (lc6 in Table 2) that appear very similar to those commonly found in calcite veins to which the reviewer refers to (see also new Supplementary figure S9). Although these veins are not the main focus of the paper because they are clearly younger and unrelated to listvenite formation, we add a brief paragraph with descriptive details for completion, with reference to similar oscillatory zoning patterns in calcite (new results section 4.2.7).

L. 501-513: "A more feasible explanation is that the zoned parts of the carbonate veins formed along a preexisting fracture or vein set." I agree, but is there any petrographic evidence supporting this in addition to the later in this section mentioned parallel sets of serpentine veins? Are carbonate and preserved serpentine vein sets characterized by the same orientation?

Apart from the presence of parallel serpentine vein sets and the observation that they can have a substantially higher porosity than matrix serpentine (which may favour preferential carbonation), we do not have clear petrographic evidence. Unfortunately, the cores are unoriented and do not permit systematic investigation of vein orientations. Therefore, we formulate this model as a hypothesis that we deem the most feasible explanation, but cannot ultimately prove it.

This section also reveals another general issue; often a model or idea is presented, but the observation itself (i.e., the evidence or indication) is described afterwards. In order to increase the comprehensibility of the authors' ideas, the observation should be mentioned first and then discussed. This also applies to other sections (e.g., discussion on crystallization pressure in chapter 5.5). See general comment (b).

We re-structured this section so that the observation of subparallel serpentine veins is mentioned earlier. Other than that we believe that the section is clearly structured, starting with a key observation followed by a discussion of two hypotheses how these may have formed.

L. 515-516: "Listvenites are inferred to form, among other settings, at the base of obducted ophiolites [...]"; does this mean that listvenites form when the ophiolite is already obducted, i.e., emplaced on continental crust or uplifted above sea level, respectively?

The exact tectonic setting and timing of listvenite formation in the Samail ophiolite is not ultimately known. Based on the most recent data, formation coeval with subduction and/or ophiolite emplacement appears to be the most likely and consistent scenario (Kelemen et al., 2022), but a different setting may also be possible. Seawater can be excluded as the fluid source because isotopic

analysis points to deep metamorphic fluids (see de Obeso et al., 2022). Because dating of ultramafic rocks (including carbonated ultramafics) is notoriously difficult, a similar uncertainty about the tectonic setting is also common for other listvenite occurrences worldwide. We briefly discuss this now in the newly added section 5.6.

*L.* 530: "[...] while the conversion of serpentine to magnesite and quartz is predicted to cause a solid volume expansion of 18 - 22 %"; is there a citation for these numbers?

**References added**

L. 534-535: What is the "chemical evidence"? Do I understand correctly that the inferred fluid film between vein minerals and wall rock argues against force of crystallization? Was the fluid film consistently existent throughout veining?

To clarify the reasoning we modified this line and the following to: "However, several observations argue against this mechanism dominating during early carbonation: (i) passive markers show that much of the vein width was accommodated by replacement rather than opening (Fig. 8), (ii) euhedral growth patterns point to the presence of an open fluid conduit at the vein-matrix interface (Fig. 11), and (iii) most zoned carbonate veins in serpentinites and listvenites contain a much smaller proportion of SiO2 (mostly as inclusions in magnesite) than expected for isochemical replacement of serpentine (Fig. 4, Fig. 7, Fig. 8), indicating that silica was leached (reaction R3)."

The main argument against force of crystallization here is point (iii), because crystallization pressure is correlated with volumetric expansion of the product phase. However, if there is substantial SiO2 leaching from the local site of magnesite precipitation during vein growth (as evidenced by the absence or scarcity of quartz within these veins), the replacement of serpentine by magnesite is not volume-increasing.

Whether the fluid film was consistently existent throughout vein growth is uncertain. However, the observed growth by replacement with progressive euhedral overgrowth followed by dendritic precipitation most likely requires a progressively renewed fluid film.

L. 538: "On the other hand" implies that the following phrase contradicts the preceding one. But, as I understand, it is an additional argument for leaching.

**Yes. We remove this expression.**

L. 540-541: "Combined influx of  $CO_2$  and local leaching of silica would thus have resulted in a solid volume decrease at the vein-serpentine interface because magnesite has a higher density than serpentine."; does this also apply if serpentine did not completely convert into magnesite, i.e., if there are further reaction products. Is there any petrographic support? L. 557: Can you explain in greater detail how quartz occurrence and expansion are related?

As noted by both reviewers, the description of chemical reactions and the related discussion of volume changes during replacement veining has not been very clear. For clarification, we added a paragraph to section 5.1. explaining the key reactions that we consider relevant for the formation of listvenites and veins in BT1B. The added reaction R3 in the revised manuscript pertains to the above reviewer's comment:

 $Mg_3Si_2(OH)_4 + 3CO_{2(aq)} = 3MgCO_3 + (2 - n)SiO_2 + nSiO_{2(aq)} + 2H_2O$

We write now in section 5.1:

"Isochemical replacement of serpentine by magnesite and quartz (n = 0 in R3) would lead to a magnesite/quartz proportion of 1.5 molar, equivalent to ~34 vol% quartz. Local mobility of aqueous silica is inferred from the observation that many matrix domains and magnesite veins have magnesite/quartz proportions significantly higher than 1.5 molar (Fig. 3; Fig. 4). Thus, only some of the released silica precipitated in-situ, forming SiO2 inclusions in magnesite or, rarely, talc (e.g., Fig. 4), indicating local leaching of SiO2,aq."

And in section 5.5:

"[...] most zoned carbonate veins in serpentinites and listvenites contain a much smaller proportion of SiO2 (mostly as inclusions in magnesite) than expected for isochemical replacement of serpentine (Fig. 4, Fig. 7, Fig. 8), indicating that silica was leached (reaction R3). Mg isotope geochemistry and bulk chemistry mass balance calculations suggest that Mg in listvenite magnesite is derived from local dissolution of the peridotite protolith (de Obeso et al., 2021; Godard et al., 2021). Assuming that external Mg influx was negligible and that magnesite growth is rate-limited by serpentine dissolution, reaction R3 would be related to solid volume expansion only if at least ~22 vol% of the solid reaction products is quartz. If more SiO2, aq is leached than precipitated insitu as quartz (n > 1 in R3), solid volume change would be negative. Since in-situ quartz abundance in zoned carbonate veins is typically < 10 vol%, combined influx of CO2 and local leaching of silica could thus have resulted in a solid volume decrease at the vein-serpentine interface because magnesite has a higher density than serpentine."

L. 572: "[...] point to an important role of tectonic stress [...]"; how does veining fit into the regional tectonic framework? Is there any additional evidence such as vein orientations in accordance with the regional stress regime at the time of formation? How can the absolute timing of vein formation be roughly constrained?

We add the following statement to the discussion:

"CO2 fluid flux derived from subduction/underthrusting of (meta)sediments below the Samail ophiolite is considered the most likely setting of listvenite formation at Site BT1 (de Obeso et al., 2022; Kelemen et al., 2022), although carbonation during extensional reactivation of the thrust fault in an early phase after obduction is possible (c.f. section 2.2). While the common parallelism of zoned magnesite veins points to a strong influence of tectonic stress on vein formation, our results do not allow to determine whether veining occurred in an overall contractional or extensional setting. This caveat is due to unoriented drill cores and the complexity arising from the observation that most of the syn-carbonation veins are replacement and not purely dilatant veins. Folding (Kelemen et al., 2022) and several phases of post-listvenite brittle faulting (Menzel et al., 2020) further complicate a reconstruction of paleo-stress directions during formation of the different vein generations."

Constraining the absolute timing of formation of the different vein generations is challenging and a considerable undertaking, and may prove impossible due to the low U/Pb concentrations in (carbonated) ultramafics. Regardless, U/Pb dating of magnesite is beyond the scope of our study.

**Figure 1: Consider to include sample points in your lithological column.**

As the used samples (overall > 100 samples, with about 30 investigated in more detail) are spread over the upper 200 m of Hole BT1B we deem it of limited use to plot their position in the small lithological column in figure 1, because it would be impractical to label single samples there. The precise sample provenance (both with respect to core sections and depth in Hole BT1B) is documented in Supplementary Table S1.

Figure 2: The zoning of the carbonate vein is difficult to identify in this figure.

We add a BSE image where the zoning is better visible as a supplementary Figure S1.

Figure 3: Add a scale to a and b.

done

Figure 5: I miss a legend indicating the Mg and Si concentrations in the maps. Also abbreviations are not explained.

The colorscale for Mg and Si concentrations is shown at the bottom of Fig. 5 f. Because mapping was done by energy-dispersive X-ray spectroscopy (EDX, see methods) without calibration on

reference materials, we cannot provide absolute concentrations. To clarify, we add a reference to the common color scale bar below f in the figure caption. Abbreviations are explained in the methods section (BSE, EDX), and abbreviations for minerals and vein generations are explained in Table 1 & 2. We add to the figure caption of Fig. 2: "For abbreviations of minerals and vein generations used in all figures, see Tables 1 & 2."

Figure 10: This figure is important for the understanding. Something like this with more focus on the respective vein types would be helpful in the results chapter.

Please see our reply to main comment (a).

Furthermore, can you give more information on the cataclastic and brecciated samples, preferentially in the main text? Give the shape of the fragments and their orientation some indication on the type of fracturing?

The listvenite cataclasites are described in detail in Menzel et al. (2020, JGR Solid Earth). Because they are generally younger than listvenite formation, we do not discuss here their relation to some of the younger veins.

The abbreviation lc is not defined.

Abbreviations for vein generations are explained in Table 1 & 2

*Table 1 and 2: Consider to have the same structure in both tables; first row: serpentinite and listvenite, respectively. Also consider to indicate the origin of the samples, drill core and fieldwork.*

First row of table 2 removed so that both tables have the same structure. We report in this table only some samples of Hole BT1B (indicated by the sample names) as examples where these vein generations are clearly visible, a full list would be too long and confusing. A more detailed description and listing of different vein generations per sample is included in supplementary Table S1.

*Clear serpentine = transparent serpentine?*

Yes, modified.

**Technical corrections**

*L. 111: Consider to change "normal to strike-slip faults" to "normal and strike slip faulting"; see also L. 424*

We prefer the formulation as it is in lines 111 and 424.

L. 137: "Veins per meter" and "veins/m"; check for consistency.

Modified to veins per meter in both cases

L. 140 and L. 143: "carbonate oxide" and "Carbonate-oxide"; check for consistency.

corrected

L. 450: "micron"

corrected

L. 517: Consider "fracturing" instead of "fracture"

modified

L. 534-535: "However [...]"; word(s) missing/incomplete phrase

To clarify the reasoning we modified lines 534 and following.

**RC2 (anonymous)**

**General comments:**

In this manuscript carbonate-veined serpentinites and listvenites from the Samail ophiolite (Oman) are described using detailed mineralogical and petrographic observations, dominantly microscopy, CL imaging and SEM. Numerous generations of serpentine-, carbonate-, and quartz-veining are described in a lot of detail resolving the evolution of the carbonation sequence and discussing the mechanisms of carbonation by mineral replacement. This manuscript is very well written and well structured. It contains a very detailed results section presenting the petrographic observations of a large range of samples from the Oman drill core. Despite the large data set, the text can be followed well and distinctions between different groups of vein formation are clearly highlighted. Hence, the data is overall clearly presented and of high quality. There are a few suggestions concerning the mineralogy and the drivers for mineral carbonation that should be considered before acceptance of this manuscript:

1. Were the different carbonate minerals only determined using the SEM? Or did you use any other techniques such as Raman spectroscopy or XRD? You mention that magnesite and dolomite were detected, but previous studies have also described the abundance of aragonite and calcite vein generations (see e.g., Ternieten et al., 2021, JGR). I would suggest determining the mineralogy of some of these veins e.g., by Raman spectroscopy for confirmation.

In this study we only used EDX-SEM to distinguish different carbonates, because carbonate in the BT1B cores is mostly magnesite (with variable Fe-content) and minor dolomite; CaCO3 is extremely rare and only occurs in very late veins in BT1B. EDX-SEM mapping is in our opinion the best method in this case to identify different carbonates and their spatial distribution, but we agree that if CaCO3 was more abundant Raman spectroscopy or XRD would be very useful.

Furthermore, I suggest to add a short summary in what way these vein generations differ or are similar to those described in e.g., Ternieten et al. (2021) (or other studies on the carbonates from the Oman ophiolite), which were done on similar drill core samples from the Oman drilling project.

*Ternieten et al.* (2021) *describe veins that are much younger than the ones that are the focus of our study.*

We write in section 4.1.5: "Late, partially open or brecciated carbonate veins cut serpentinites and all previous vein generations (sc4, Table 1). These are unrelated to the formation of listvenite, and possibly linked to young magnesite, dolomite and calcite/aragonite precipitation in open joints from groundwater or hyperalkaline serpentinization fluids. Similar young carbonate veins and travertine are common in the weathering horizon of the Samail ophiolite peridotites (e.g., Chavagnac et al., 2013; Giampouras et al., 2020; Noël et al., 2018; Ternieten et al., 2021)."

And we add in section 5.1: "Some of the youngest, post-listvenite magnesite and dolomite vein generations (lc5, lc6) may have a similar origin as very young magnesite, dolomite and calcite/aragonite veins related to interaction of Mg-HCO3– and Ca-OH– groundwater with the Samail peridotite (Noël et al., 2018; Streit et al., 2012; Ternieten et al., 2021)"

2. Furthermore, is there any change in mineralogy within these carbonate veins from early to later formation (i.e., vein generation)? And can you provide any information in what way the fluid conditions would have favoured the formation of magnesite over dolomite and vice-versa? Generally, how did factors (as those for example mentioned at the beginning of section 5.2) control the precipitation of magnesite versus dolomite versus potentially calcite or aragonite that occur in some of these drill cores?

Many early veins are composed mostly of magnesite, and dolomite usually occurs in later veins. Thus there seems to be a broad general trend from early magnesite formed by local reaction with CO2, aq to a larger influence of dissolved Ca2+, Mg2+, (bi)carbonate and SiO2, aq aqueous species later on, which may correlate with progressive cooling. This trend is however a somewhat simplified interpretation as there are a number of exceptions: some dolomite also occurs in early veins (either as zones or as alternating segments of otherwise magnesite veins, see section 4.1.3), while there are also late magnesite veins (lc5 veins). We have added more explanations and discussion about this question in section 5.1 (see text explaining reactions R2 - R4).

We also add in the new section 5.6: "With subsequent progressive cooling of the ophiolite and underlying units, CO2 concentration in infiltrating fluids likely decreased. Thus, the fluid chemistry may have switched to more Ca- and/or Mg- bicarbonate ionic solutions favoring the formation of syntaxial, post-listvenite carbonate veins, in contrast to the earlier syn-listvenite replacement veins that formed by reaction of serpentine with CO2."

3. Finally, it would also be useful to state how the mineral replacement from peridotite to a magnesite-quartz rock proceeds. You mention in the discussion that carbonation proceeds via mineral dissolution. What would drive mineral dissolution and replacement in these veins (see e.g., in lines 474/475)? Can you infer the conditions that would have driven mineral replacement rather than simple filling of fractures? It would be good to further expand on these points.

Thank you for this comment –we realize that it would be useful to provide a bit more background about the chemical reactions involved in carbonation in the discussion, rather than only referring to literature. We added a paragraph to section 5.1 describing the key reactions of replacement, which are primarily driven by the instability of olivine, brucite and serpentine in the presence of high  $CO_2$  concentrations in fluid. In veins that form (or widen) via replacement,  $CO_{2,aq}$  is inferred to be the main driving force of serpentine dissolution coupled with magnesite-quartz precipitation, whereas post-listvenite syntaxial and late veins are inferred to have formed primarily as fracture infills precipitated from cation- and (bi)carbonate bearing solutions.

**Specific comments:**

*Line 15: It would be better to specify throughout the text if you are talking about magnesite and/or dolomite veins.*

Yes, that is true. We correct carbonate to "magnesite" where appropriate

Line 19-20: Same as above, it would be better to specify what carbonate minerals make up the veins, since dolomite is also a carbonate mineral.

We correct carbonate to "magnesite"

Line 33: add the meaning of the abbreviation of IPCC.

done

Line 72: You can already add here a reference to Fig. 1

**Added in line 71**

Line 202: To me these magnesite veins look rather random than following the serpentine mesh texture. It might be better to use a thin section image here rather than a BSE image where the mesh texture is not visible.

We prefer to show here a BSE image because that also shows the chemical zoning in magnesite. We added an additional reflected light image as supplementary figure S1b, which shows magnesite along the polygonal outlines of serpentine mesh.

*Line 255: "Sq1" if It would be useful to label these types of veins in Fig. 5a.*

Labels added in Fig. 5

*Line 257: Was the resolution of the EDS maps high enough to reveal nanometer-sized mineral inclusions? What is shown in the figures is all only resolvable to the micrometer scale.*

It is correct that the resolution of the shown EDS map is not high enough to resolve single nm-sized inclusions, only to show that the impurities in quartz occur at a sub-micron scale (and thus, in the range of nm to µm scale inclusions). It is possible that this impure quartz has dispersed Mg incorporated in its crystal lattice, but from the knowledge from previous TEM studies (Beinlich et al., 2020; Menzel et al., 2021) that SiO2 nano-inclusions are the cause for apparently silica-enriched magnesite we consider it very likely that it is similar in this case of Mg-enriched quartz.

*Line* 278: *Is there any theory why the magnesites follow the mesh rims, but the center of the serpentinite-mesh texture was replaced by quartz?*

We have been wondering about this question, and don't know the reason yet. It probably comes down to different fluid flow rates in these distinct microenvironments with differing porosity and reactivity, which cause chemical gradients that favour replacement by quartz in the mesh centers (and in bastite).

Line 313: What is the evidence that these are silica nano-inclusions? Is this only based on elevated Si contents in the EDX maps? Or did you detect them as individual mineral phases using the FE-SEM? Fig. 7f only shows an overview BSE image, but does not allow the identification of nanoscale mineral phases.

It is correct that this is not visible in Fig. 7f. That these are mostly SiO2 nano-inclusions (and not dispersed silica in the carbonate crystal lattice) is known from previous investigations of magnesite in the listvenite matrix by scanning transmission electron microscopy (STEM). We add the following remark in this line: "Similar silica inclusions are common in matrix magnesite of the listvenite, where they have been shown to be mostly SiO2 nano-inclusions (Beinlich et al., 2020; Menzel et al., 2021)."

Line 369: might be simpler to just call this "microcrystalline quartz" rather than chalcedony.

We use the term chalcedony here because it is the common term to describe certain microcrystalline quartz types with fibrous / radial habits. The fibrous to radial chalcedony aggregates in these quartz/chalcedony-magnesite veins are rather different from the massive microcrystalline quartz described in the section before.

Line 386: it would be good to label the panels in Fig. 10 with a,b,c etc. and then refer to them when discussing the different stage of rock formation below.

We have considered this suggestion, but we think this may add confusion; in our opinion it is more important here to which stage I - X the panels correspond. We realize that these panel labels may have been easy to overlook, therefore we modify the figure to make them bold.

Line 392: Is there a reference for the serpentinization temperature?

The serpentinization temperature is not well constrained yet for the BT1 serpentinites. We write "likely at T < 250 °C", because most serpentine is lizardite, chrysotile and/or polygonal, while antigorite is uncommon (see Kelemen et al., 2022).

*line 400: "ellipsoidal/spheroidal grains" if Are these single grains or mineral aggregates? Typically, single carbonate grains are not ellipsoidal when they precipitate.*

Yes, these are single grains (see Beinlich et al., 2020b; Menzel et al., 2021). As described in the cited papers, their rather special habit is interpreted to be a result of disequilibrium precipitation at high oversaturation under deviatoric stress (when ellipsoidal).

Line 401: specify in which panel this is seen: Fig. 10b?

See above reply

Line 426: It would be useful to have the sequence of reactions that take place during serpentinite replacement written out somewhere, such as:

 $Mg3Si2O5(OH)4 + \hat{a} \square \exists 3CO2, aq \rightarrow 3MgCO3 + \hat{a} \square \exists 2SiO2, aq\hat{a} \square \exists \hat{a} \square \exists 2H2O$

Good remark, including the reactions will make the discussion more accessible. We add a paragraph describing the key chemical reaction to section 5.1.

Line 427: "steps (4) - (8) if Do you mean here the steps described above? If yes, it would be useful to use roman numbers here.

Yes. done

*Line 436: This should be Schwarzenbach et al., 2016 (please also adjust the reference list)*

corrected

*Line 449: Did you find any evidence for nano-porosity in these samples when studying them with the SEM?*

Yes. We added a figure obtained by FE-SEM of a broad ion beam polished sample (Fig. 12), and we update the methods section accordingly.

Line 507: Did you determine if the serpentine veins, that are partly replaced by carbonates, are either chrysotile or lizardite, e.g., using Raman spectroscopy?

No, we have not used Raman spectroscopy here. A detailed characterization of the different serpentine polytypes (lizardite, chrysotile; but also polygonal serpentine, nanotubular vs non-tubular varieties, proto-lizardite/chrysotile, intergrowths between different types or with brucite) and their spatial distribution in the serpentinite protolith would be very useful to better understand the carbonation processes. Such an analysis that might include Raman spectroscopy but also other methods is however a significant undertaking and beyond the scope of this paper.

*Fig 2 (line 855): In what way are these carbonate veins pseudomorphic? Pseudomorphic after which mineral phase? Please specify.*

Pseudomorphic after mesh-textured serpentine and after serpentine crack-seal veins. We modify the caption of figure 2 to clarify this. Please find this information also in the text (sections 4.1.2 and 4.2.1).

Line 863: "partial replacement by magnesite and crosscut by zoned carbonate veins" if are the carbonate veins not also magnesite? Or are there any other carbonate mineral present in these veins?

The sc2 veins in this sample are rimmed by dolomite in addition to magnesite; we modify the figure caption to clarify that these are zoned magnesite-dolomite veins here.

*Fig. 4 (line 873): Which elements where measured in this thin section and are shown in the composite-color EDS maps?*

We added an indication of the elements that make up the primary colors red, blue, cyan and yellow in this composite image.

*Fig.* 4g: What is shown by the yellow arrow? Please add this to the figure caption.

It shows euhedral dolomite at the vein margins; we add mention of the arrow in the caption.

*Fig. 5 (line 930): Please label the carbonate and quartz in Fig. 5d and e. What does "ViP xpol" mean?*

Labels added. ViP stands for Virtual polarizing microscopy; we modified the methods section to explain this abbreviation there.

---

## Author Response (AR2)

**Comments by topical editor Virginia Toy**

**Comments to the author**:
I have reviewed the revisions you made in response to the previous reviews, and find them reasonable.

I did identify the following further minor revisions I think you should make prior to acceptance for publication;
- L423 Perhaps you should paraphrase the response to reviewer 2 about the origin of the ellipsoidal grains you describe?
- L482 'have a similar origin to'
- L648 'do not allow us to determine'
- Please label the porosity in new Fig. 12. Also, you use this figure to demonstrate the higher porosity in the fibrous serpentine veins than matrix serpentine - in that case I think you should add a panel that shows the matrix serpentine for comparison.
- Please incorporate the new figures you added to the supplement in response to the Reviewer's comments into the main manuscript - I don't think most people will see them if they are only in the supplement, and they are clearly of interest to readers (and reviewers).

**Author response**
Dear Virginia, dear editors,
thank you for your remarks. In the revised manuscript version, we have:
- Included a paraphrased version of our response to a comment by reviewer 2 in L423: „The ellipsoidal/spheroidal grain habit is interpreted to be a result of disequilibrium precipitation at high oversaturation (Beinlich et al., 2020b) and/or under deviatoric stress (Menzel et al., 2021)."
- corrected L482 as suggested
- corrected L648 as suggested
- labelled the porosity in Fig. 12 and added a second panel to this figure where the porosity in matrix serpentine in contrast to the vein is better visible
- incorporated two images in Fig. 2 and one image in Fig. 9 that we before had added as supplementary figures in response to reviewer's comments. We updated figure references throughout the text accordingly.

We thank again all reviewers and editors for their input which helped to improve the manuscript, and hope that is now suitable for publication.

With kind regards on behalf of all authors,
Manuel Menzel
(corresponding author)

May 6, 2022